# Novel Therapeutic Approaches in Pediatric Acute Lymphoblastic Leukemia

**DOI:** 10.3390/ijms262311362

**Published:** 2025-11-24

**Authors:** Maria Maddalena Marrapodi, Alessandra Di Paola, Giuseppe Di Feo, Oriana Di Domenico, Martina Di Martino, Lucia Argenziano, Marianna Falcone, Daniela Di Pinto, Francesca Rossi, Elvira Pota

**Affiliations:** 1Department of Woman, Child and General and Specialist Surgery, University of Campania “Luigi Vanvitelli”, 80138 Napoli, Italy; mariamaddalena.marrapodi@unicampania.it (M.M.M.); or a.dipaola@unilink.it (A.D.P.); giuseppe.difeo.bio@gmail.com (G.D.F.); didomenico.oriana.odd@gmail.com (O.D.D.); martina.dimartino@policliniconapoli.it (M.D.M.); lucia.argenziano@unicampania.it (L.A.); mariannafalcone.1985@libero.it (M.F.); daniela.dipinto@policliniconapoli.it (D.D.P.); elvira.pota@gmail.com (E.P.); 2Department of Life Science, Health and Health Professions, Link Campus University, 00165 Rome, Italy

**Keywords:** acute lymphoblastic leukemia, B-ALL, T-ALL, conventional therapy, target therapy, immunotherapy

## Abstract

Acute lymphoblastic leukemia (ALL) is the most common pediatric malignancy, characterized by the clonal proliferation of immature lymphoid precursors. The distinction between B-cell ALL (B-ALL) and T-cell ALL (T-ALL) is fundamental, as each subtype exhibits distinct cytomorphological, genetic, and clinical features influencing prognosis and therapeutic strategies. Conventional multi-phase chemotherapy has significantly improved survival rates, yet its efficacy is limited by severe short- and long-term toxicities, highlighting the need for more selective therapeutic approaches. Advances in molecular profiling have enabled the identification of key oncogenic pathways, paving the way for targeted therapies such as tyrosine kinase inhibitors (TKIs), JAK-STAT pathway inhibitors, BCL-2 antagonists, and agents modulating epigenetic and cell cycle regulators. Concurrently, immunotherapeutic strategies have transformed the therapeutic landscape of pediatric ALL. Bispecific antibodies such as blinatumomab (anti-CD19), antibody–drug conjugates like inotuzumab ozogamicin (anti-CD22), and monoclonal antibodies such as daratumumab (anti-CD38) have demonstrated efficacy in relapsed or refractory disease with improved safety profiles. Moreover, CAR-T-cell therapy, particularly CD19-directed products, has shown unprecedented remission rates in refractory B-ALL. The integration of targeted and immune-based therapies into conventional regimens represents a decisive step toward precision medicine, aiming to enhance survival outcomes while reducing treatment-related toxicity and improving quality of life in ALL children. This review aims to provide a comprehensive overview of the current understanding of ALL pathobiology and therapeutic approaches, with particular emphasis on the expanding role of immunotherapeutic strategies in pediatric disease.

## 1. Introduction

Acute lymphoblastic leukemia (ALL) represents the most common malignancy in children and adolescents, with an incidence of approximately 3 cases per 100,000 individuals per year. It is a severe hematologic neoplastic disease caused by the clonal proliferation and accumulation of immature lymphoid progenitor cells (lymphoblasts) in the bone marrow, peripheral blood, and, in some cases, extramedullary sites such as the central nervous system (CNS) and lymph nodes [1,2,3,4,5]. These lymphoblasts fail to differentiate properly and expand rapidly, becoming malignant cells that impair normal hematopoiesis and ultimately cause bone marrow failure [1,4]. ALL is classified based on the lymphoid lineage involved. The most common form is precursor B-cell ALL (BCP-ALL), accounting for approximately 80–85% of cases. T-cell ALL occurs in 10–15% of cases, while mature B-cell ALL is rare, representing less than 5% of cases [6]. Clinical manifestations are mainly due to the accumulation of immature lymphoid cells in the bone marrow and other tissues. Symptoms are often non-specific and include fatigue, fever, weight loss, and back pain, alongside signs of bone marrow failure, such as anemia, infections, and easy bruising or bleeding. Around 20% of patients show extramedullary involvement, including lymphadenopathy and hepatosplenomegaly [4,7]. CNS involvement is seen in 5–8% of cases at diagnosis and may present with cranial nerve dysfunction or meningeal signs [4,8].

ALL is characterized by significant heterogeneity in its clinical and biological features, driven by the acquisition of various genetic alterations in leukemic clones [3,5]. B-ALL, the most common form of ALL, consists of more than 20 molecularly distinct subtypes, each showing age-related prevalence and specific genetic expression profiles. There are three key types of genetic alterations: chromosomal aneuploidies, gene rearrangements that either lead to oncogene deregulation or generate fusion transcription factors, and point mutations [1]. Each B-ALL subtype is further characterized by additional mutations that are responsible for the dysregulation of key pathways involved in lymphoid differentiation, cell cycle control, kinase signaling, and chromatin remodeling.

T-ALL results from the accumulation of genetic alterations that disrupt normal mechanisms regulating cell growth, differentiation, proliferation, and survival during thymic development. The disease exhibits marked genetic heterogeneity, with chromosomal abnormalities identified in the vast majority of cases.

In ALL, genetic risk is further influenced by rare syndromes and familial cancer syndromes such as Li-Fraumeni, as well as non-coding DNA polymorphisms. Down syndrome is strongly associated with an increased risk of B-ALL, while ataxia–telangiectasia is linked to T-ALL [4,9].

Environmental and infectious exposures have also been implicated in ALL development. Reported factors include exposure to toxic solvents, pesticides, ionizing radiation, and viral agents such as Epstein–Barr Virus (EBV) and Human Immunodeficiency Virus (HIV) [1,4,10]. Nevertheless, in the vast majority of cases, ALL arises de novo in previously healthy individuals.

Currently, leukemia accounts for more than 25% of all cancers diagnosed in individuals under 20 years of age, with incidence rates gradually increasing. Importantly, survival outcomes have markedly improved: the five-year survival rate has risen from below 50% in the 1970s to nearly 85% today, thanks to advances in risk stratification and treatment approaches [6].

Nowadays, the evaluation of genetic alterations in ALL plays a pivotal role in therapeutic interventions and pediatric patients’ prognosis. Moreover, to ensure the customization of treatment for each patient, it is necessary to take into consideration patients’ response to treatment based on minimal residual disease (MRD). MRD refers to a chemotherapy/radiotherapy-surviving leukemia cell population that gives rise to disease relapse. MRD detection is critical for predicting the outcome and for selecting the intensity of further treatment strategies. Current methods used to assess MRD are based on evaluations of phenotypic markers or differential gene patterns through polymerase chain reaction (PCR), real-time quantitative polymerase chain reaction (RQ-PCR), reverse transcription polymerase chain reaction (RT-PCR), or flowcytometry, together with the development of various new diagnostic platforms—including next-generation sequencing (NGS) [11].

This review aims to provide an overview of the current understanding of the pathobiology and therapeutic landscape of ALL, with a particular focus on the growing role of immunotherapeutic strategies. These novel treatments have not only improved treatment responses but have also significantly enhanced patients’ quality of life.

## 2. Genetic Alterations in Acute Lymphoblastic Leukemia

The past decade has witnessed substantial progress in elucidating the molecular and genetic underpinnings of leukemogenesis and therapeutic response in ALL. Both B-cell and T-cell lineages encompass multiple molecular subtypes, defined by distinct profiles of somatic structural alterations and sequence variants [12]. These genetic lesions frequently disrupt key biological processes, including lymphoid lineage specification, cytokine receptor signaling, kinase and Ras pathway activation, tumor suppressor gene function, and chromatin remodeling. Furthermore, recent studies have provided critical insights into the genetic architecture of clonal evolution in the development and relapse of disease, as well as the contribution of germline variants to leukemogenic susceptibility [12]. ALL arises from either B-cell precursor (BCP-ALL) or, less frequently, T-cell precursor (TCP-ALL) lineages. Both entities encompass heterogeneous subtypes, typically characterized by initiating structural chromosomal alterations. These primary lesions are often accompanied by secondary somatic events, including DNA copy number variations and sequence mutations, which collectively drive leukemogenesis. The chromosomal abnormalities observed in ALL include aneuploidy and rearrangements that lead to oncogene dysregulation or the formation of chimeric fusion transcripts with oncogenic potential [12].

### 2.1. Genetic Alterations in B-Cell Acute Lymphoblastic Leukemia

B-ALL represents the most prevalent subtype of ALL, with over twenty distinct subtypes, whose prevalence varies with patient age. These subtypes are characterized by unique gene expression signatures and are primarily driven by three classes of initiating genetic events (Figure 1):Numerical alterations leading to chromosomal aneuploidy;Structural alterations leading to chromosomal rearrangements that lead to oncogene deregulation or generate chimeric transcription factors;Point mutations.

Each subtype is further defined by co-occurring genetic alterations that disrupt pathways critical for lymphoid development, cell cycle control, kinase signaling, and chromatin remodeling. The specific genes affected and their mutation frequencies differ across subtypes, and several approaches are employed to assess them. The preferred method for identifying all relevant categories of gene fusions is hierarchical Fluorescent In Situ Hybridization (FISH) screening, which is used to confirm whether a particular gene is involved. Screening for rare fusion genes can be restricted to cases in which the presence of all other primary lesions (*ETV6*, *ABL1*, *KMT2A*, or *TCF3* fusions and hyper/haplo/hypodiplody) has been excluded [13,14]. Although the respective fusion partner should be identified subsequently in all hub gene-positive cases with translocation-specific dual FISH probes, the preferred method for simultaneously capturing all relevant small- and large-scale quantitative changes is the DNA array analysis. The combined approach of probes and an array can be viewed as an extended and refined form of standard cytogenetics [15].

Regarding the stratification of patients, first of all, they have to be investigated for the presence of BCR-ABL1. BCR-ABL1-positive (Philadelphia chromosome-positive, Ph+) ALL, arising from the translocation (t9;22), accounts for approximately 2–5% of childhood and 25% of adult ALL cases. Although historically linked to poor outcomes, the introduction of TKIs has significantly improved prognosis in this subgroup.

To allow for the identification of kinase-activating genetic alterations potentially targetable by TKIs, several other translocations (*PDGFRB*, *ABL1*, *ABL2*, and *CSF1R*, as well *CRLF2*, *IGH*, *EPOR*, and *NTRK3*) are analyzed in all BCP-ALL patients with positive MRD results on day 33 from the beginning of therapy (TP1) and without positive results for *ETV6-RUNX1*, rearranged *TCF3*, and hypodiploidy [10].

Another BCP-ALL subgroup is represented by BCR-ABL1-like leukemia, which presents a transcriptional profile similar to that of BCR-ABL-positive ALL cases; despite lacking the *BCR-ABL1* transcript, they retain the associated poor prognosis [10].

Moreover, all patients should be screened for *KMT2A* (*MLL*) rearrangements, and the fusion partner should be identified in *KMT2A*-positive cases. This alteration is especially common in patients younger than 1 year of age, and, in particular, infant pro-B leukemia with any *MLL-AF4* arrangement is associated with poor prognosis.

Other relevant gene alterations that should be assessed in children with B-precursor ALL are *TCF3-HLF* (or *E2A-HLF*), *TCF3-PBX1* (or *E2A-PBX1*), and *ETV6-RUNX1* (or *TEL-AML1*), and, most importantly, hypodiplody should be ascertained in all BCP-ALL patients. These are translocations that join two genes to form a chimeric protein that is involved in functions different from the original ones. *ETV6-RUNXI*, which results from the t(12;21) translocation, is observed in one-quarter of children with ALL and is associated with a favorable prognosis, whereas *TCF3-HLF*—resulting from the translocation (t17;19)—is rare (<1% of children with BCP-ALL) and prognostically extremely unfavorable [12].

Recently, mutations in genes regulating lymphoid development and alterations in the *IKZF1* gene encoding the lymphoid transcription factor IKAROS have been identified, and they have been given a negative prognostic value. In fact, after patient screening, *IKFZ1* deletions are found in almost 7% of ALL cases. Moreover, co-occurring deletions in *CDKN2A*, *CDKNA2B*, *PAX5*, or *PAR1* are grouped as IKFZ1plus and confer the worst outcome combined with the persistence of MRD positivity at TP1 and on day 78 from the beginning of therapy (TP2). The single aberrations that define the IKFZ1plus pattern can be detected in combination with other recurrent genetic alterations of known prognostic importance—such as *ETV6-RUNX1* (*TEL-AML-1*), *TCF3-PBX1* (*E2A-PBX1*), *TCF3-HLF* (*E2A-HLF*), and *KMT2A* (*MLL*) rearrangements or hypodiploidy. Therefore, IKFZ1plus qualifies patients for high-risk treatment if they are not PCR-MRD-negative at TP1 and not found to be positive for *ETV6-RUNX1* (*TEL-AML1*), *TCF3-PBX1* (*E2A-PBX1*), or *KMT2A* (*MLL*) rearrangements other than *KMT2A-AFF1 (MLL-AF4*) and *TCF3-HLF* (*E2A-HLF*) fusion or hypodiploidy [10,16].

Another group of gene alterations with an impact on prognosis in children with ALL is aneuploidies. Ploidy can be determined by using genetic methods such as metaphase karyotyping and DNA array analyses or by measuring the DNA content with flow cytometry. Flow cytometry expresses the DNA content as the DNA index (DI), which is the ratio between the normal amount of fluorescence seen in a diploid cell and the fluorescence content of bone marrow blasts in G0/G1 at diagnosis. Hypodiploidy is defined by a chromosome count of less than 45 and a DI of less than 0.8 and is commonly associated with high-risk leukemia, which occurs in 2–3% of children with B-precursor ALL. Conversely, 25–30% of children with B-precursor ALL have high hyperdiploidy (>50 chromosomes), which is associated with an excellent prognosis [8].

Therefore, according to their genetic background, patients could be stratified as follows:Early high risk (early HR);High risk (HR);Standard risk (SR);Medium risk (MR).

### 2.2. Genetic Alterations in T-Cell Acute Lymphoblastic Leukemia and Early T-Cell Progenitor Acute Lymphoblastic Leukemia

In patients with TCP-ALL, specific gene alterations implicated in the neoplastic transformation process and with a prognostic and therapeutic role have been identified (Figure 2). Chromosomal translocations are pivotal events in T-ALL, frequently resulting in the juxtaposition of T-cell receptor (TCR) promoters on chromosomes 7 (*TCRB*, *TCRG*) and 14 (*TCRA*, *TCRD*) with transcription factors such as TAL1, LYL1, and HOX11, thereby inducing aberrant transcriptional programs that drive leukemogenesis [17]. Additional cytogenetic alterations in TCP-ALL include fusion genes encoding chimeric oncogenic proteins, such as SIL-TAL1 and MLL, which, in some studies, have been associated with favorable clinical outcomes [17].

Beyond these structural changes, the most recurrent mutations in T-ALL involve *NOTCH1*, a gene critical for the regulation of hematopoietic progenitor differentiation toward the T-cell lineage. Disruption of this tightly controlled pathway contributes to leukemic transformation. *NOTCH1* mutations frequently co-occur with alterations in *FBXW7*, which encodes an E3 ubiquitin ligase responsible for the degradation of activated NOTCH1. Recent evidence indicates that patients harboring *NOTCH1* and/or *FBXW7* mutations represent a subgroup with favorable prognosis, particularly when associated with rapid treatment response or MRD negativity [17,18].

High-throughput sequencing studies have further elucidated the T-ALL genetic landscape, highlighting the recurrent activation of oncogenic pathways, including IL-7R/JAK/STAT, PI3K/AKT/mTOR, and RAS/MAPK. These findings not only enhance our understanding of T-ALL biology but also support the incorporation of *NOTCH1*/*FBXW7* mutation status into contemporary risk stratification models while providing a rationale for the development of novel targeted therapeutic strategies [18].

Early T-cell precursor (ETP) ALL is a distinct subtype characterized by a low expression of T-cell surface markers (CD1a, CD8, and CD5) and an aberrant expression of myeloid or stem cell markers. ETP-ALL is associated with poor prognosis, although outcomes have improved with contemporary risk-adapted treatment strategies. Genetically, ETP-ALL is heterogeneous, harboring mutations in pathways related to hematopoietic and lymphoid development (*RUNX1*, *IKZF1*, *ETV6*, *GATA3*, and *EP300*), Ras and cytokine receptor signaling (*NRAS*, *IL7R*, *KRAS*, *JAK1*, *JAK3*, *NF1*, and *PTPN11*), and epigenetic regulators (*EZH2*, *SUZ12*, *EED*, and *SETD2*) [13]. The gene expression profile of ETP-ALL resembles that of hematopoietic stem cells, suggesting that it represents an immature leukemia within a broader spectrum rather than a conventional T-ALL [19,20,21].

## 3. Cytofluorimetric Classification of B-ALL and T-ALL

### 3.1. B-Cell Acute Lymphoblastic Leukemia

B-ALL was first immunologically classified in 1995 based on the immunophenotypic characteristics corresponding to a block in normal lymphoid maturation. These leukemic populations are typically homogeneous and can be distinguished from the continuum of antigens expressed during normal B-cell maturation [22]. In general, B-ALL is characterized by the expression of CD19 (nearly 100% of cases), CD79a, cytoplasmic CD22 (cCD22), terminal deoxynucleotidyl transferase (TdT), and HLA-DR and a dim expression of CD45 [23]. Among these, CD19 and cCD22 are considered the most sensitive markers for B-cell lineage.

The precursor nature of B-ALL is confirmed by the expression of CD34 (≈70%) and TdT (>90%), along with the absence of cytoplasmic or surface light chains, markers that tend to disappear in later maturation stages. In fact, only ~4% of cases lack both CD34 and TdT, and, notably, TdT expression in B-ALL is typically more intense than in T-ALL or AML [23].

Other antigens expressed as CD10 and cytoplasmic immunoglobulin heavy chains (cμ) further help to subclassify B-ALL. While surface immunoglobulin (sIg) indicates mature B-cell neoplasms, exceptions do exist [24].

Although CD20 is a marker of mature B cells, it is expressed in ~20–40% of B-ALL cases, highlighting a discordant antigen arrangement not seen in normal maturation; nonetheless, it is associated with poor prognosis in adults [25].

CD10, formerly called the “common ALL antigen,” is expressed in 80–90% of B-ALL cases but is not specific, as it is also seen in other B-cell malignancies (e.g., follicular lymphoma and Burkitt lymphoma). CD10 must be interpreted in the context of the full immunophenotypic and genetic profile and clinical presentation. Moreover, CD10-negative B-ALL is often linked to MLL translocations and poor prognosis (i.e., pro-B-ALL).

Although most B-ALL cases show clonal IGH rearrangements, TCR rearrangements are also seen in up to 70% of cases. Conversely, T-ALL can show IGH rearrangements in 20% of cases, suggesting that these genetic findings are not sufficient for lineage determination [26].

### 3.2. T-Cell Acute Lymphoblastic Leukemia

The precursor origin of T-ALL cells is indicated by the presence of TdT, CD1a, and/or CD34 and by the absence of surface CD3 (sCD3). Aberrant co-expression of both CD4 and CD8—or the lack of both—is also suggestive of an immature immunophenotype.

HLA-DR, although universally expressed in B-ALL, is rarely seen in T-ALL and is considered a marker of early T-cell development. A small study reported HLA-DR expression in 40–50% of pro- and pre-T-ALL cases but not in more mature T-ALL [27].

TdT is the most consistently expressed precursor marker (>90% of cases), whereas CD34 is only present in about 30–40% of cases [28,29]. These markers are more likely to be absent in mature subtypes, such as medullary T-ALL, which can lead to diagnostic confusion with mature T-cell lymphomas. Additionally, blasts that are both CD34− and HLA-DR− may still be encountered in these settings. Among the T-ALL subtypes, more mature variants tend to have a better prognosis than the early/pre-thymic types. In fact, cortical T-ALL (CD1a+) is generally associated with the most favorable outcomes across both adult and pediatric populations. CD10, though transiently expressed during normal T-cell development, is observed in 30% of T-ALL cases. However, recent studies suggest that it has minimal prognostic value under current treatment protocols. A particularly immature form, termed early T-cell progenitor (ETP) ALL, has been identified in 10–15% of childhood cases. ETP-ALL is defined by cCD3 positivity, CD1a and CD8 negativity, dim CD5, and frequent co-expression of myeloid markers (e.g., CD117, CD13, and CD33), along with other immature markers such as CD34, HLA-DR, TdT, and CD133. This subgroup has a high risk of remission failure or relapse (57% at 2 years versus 14% for non-ETP T-ALL). Consequently, intensive therapy, including allogeneic stem cell transplant, is often recommended upon diagnosis [30].

## 4. Conventional Acute Lymphoblastic Leukemia Treatments

Conventional therapy for ALL is based on a combination of different agents: hormone treatment (glucocorticoids), amino acid depletion (asparaginase), antimetabolites, alkylating agents, anthracyclines, and metaphase blockers [31,32]. In cases of Ph+ or Philadelphia chromosome-like ALL, TKIs are also administered [33].

The standard chemotherapy protocol used in Europe consists of four phases: induction, consolidation, re-induction, and maintenance [34]. Induction therapy aims to restore normal hematopoiesis and is based on the administration of steroids, vincristine, asparaginase, and anthracyclines. At the end of this phase, complete remission is observed in approximately 95% of patients, although some children may experience severe adverse events [34,35].

Induction is followed by the first phase of consolidation therapy, consisting of cytarabine and cyclophosphamide for 4 to 6 weeks depending on the patient’s risk group. Based on treatment response, patients are classified into three risk categories: standard risk (SR), intermediate risk (IR), and high risk (HR). The second phase of consolidation then begins: SR and IR patients receive high-dose methotrexate (HD-MTX) in combination with mercaptopurine, while HR T-ALL patients receive three blocks of polychemotherapy, and HR B-ALL patients receive one block of polychemotherapy, followed by two cycles of blinatumomab.

Subsequently, re-induction therapy is administered, using a drug combination similar to induction therapy [34,36]. For patients at high risk of relapses, allogeneic stem cell transplantation (allo-SCT) represents the main consolidation treatment. Patients referred for transplantation undergo preconditioning with myeloablative therapy, such as total body irradiation or busulfan, in order to eliminate leukemic cells resistant to standard chemotherapy. Despite advances in transplantation reducing both mortality and morbidity, a number of late adverse effects have been reported, including infertility, growth retardation, metabolic disorders, and secondary malignant neoplasms. Therefore, allo-SCT is recommended only for high-risk patients [34].

Finally, maintenance therapy lasts for 1–2 years and consists of daily 6-mercaptopurine (6-MP) and weekly methotrexate (MTX) therapy [34].

In fact, only patients with positive SNC at diagnosis receive cranial radiotherapy.

## 5. Toxicity Related to Conventional Therapy

Conventional therapy in ALL causes several acute side effects that affect different organs. The most recurrent side effects are opportunistic infections, mucositis, bone toxicities, central or/and peripheral neuropathy, endocrinopathies, hyperlipidemia, sinusoidal obstruction syndrome (SOS) and hepatotoxicity, thromboembolism (TE), HD-MTX-induced nephrotoxicity, asparaginase-associated hypersensitivity, allergy, and pancreatitis [37].

### 5.1. Mucositis

In total, 40% of patients experience mucositis, which is mostly caused by the administration of high-dose antimetabolites or alkylating agents. It affects both the mouth (oral mucositis) and intestines (intestinal mucositis), leading to the development of several symptoms like ulcers, pain, nausea, and diarrhea. These effects peak around 10–14 days after chemotherapy, coinciding with neutropenia [37,38].

Mucositis results from inflammation triggered by damage-associated signals and may be worsened by microbiome disruption. Severe mucositis compromises the gut barrier, increasing the risk of systemic infections. Interestingly, in many febrile neutropenic episodes, no infection is found, suggesting that systemic inflammation (with elevated CRP or IL-6) is often the main cause, leading to “febrile mucositis” [37,39].

### 5.2. Bone Toxicities

Osteoporosis (OP) and osteonecrosis (ON) represent the most significant skeletal complications of pediatric ALL therapy. The pathophysiology of OP is not completely clear, but both leukemia itself and corticosteroid therapy contribute to bone loss and increased fragility, often resulting in fractures—particularly multifocal compression fractures of the spine [40,41].

ON is caused by bone tissue death due to inadequate blood supply [42]. ON should be suspected starting from the second year of treatment, but it may also present earlier or sometimes after stop therapy. The hips and knees are most frequently affected, and multiple joint involvement is frequent in both clinical and subclinical presentations [42,43]. The clinical course can be debilitating, with patients often experiencing daily pain, reduced physical activity, and impaired quality of life. In severe cases, children may become dependent on wheelchairs. Long-term consequences may include articular surface collapse, progressive joint damage, and a need for surgical interventions.

The pathogenesis of ON appears to be multifactorial. Corticosteroids are a key contributor since they can induce hypertrophy of bone marrow adipocytes, increasing intraosseous pressure and compromising the bloodstream. Additionally, corticosteroids may be directly toxic to osteocytes. Other proposed mechanisms include the formation of fat emboli, vasculitis, or microvascular thromboses that further disrupt the vascular supply [44]. Hyperlipidemia, a common side effect of corticosteroids and asparaginase treatments, seems to be a contributing factor to ON [37]. The administration of dexamethasone in an intermittent schedule is the only preventive strategy to reduce the risk of ON [45].

### 5.3. Neurotoxicities

CNS toxicities affect 10–15% of children treated for ALL, causing several syndromes such as seizures [46], MTX-induced stroke-like syndrome (MTX-SLS) [37], posterior reversible encephalopathy syndrome (PRES) [47], and steroid-induced psychosis [37]. These syndromes can contribute to persistent neurocognitive impairments—including deficits in attention and executive function [48]. Corticosteroids may exert neurotoxic effects through modulation of neurotransmitters and impairment of the hypothalamic–pituitary–adrenal axis [49]. Genetic polymorphisms affecting neurogenesis and drug metabolism have been suggested as risk factors but require further validation [50]. Seizures (in approximately 10% of pediatric ALL patients) may present in isolation or alongside other neurologic events (e.g., PRES and MTX-SLS) or result from metabolic, infectious, or vascular complications, with female sex emerging as a risk factor [51]. MTX-SLS typically arises 2–14 days after HD or intrathecal MTX, causing focal neurological symptoms such as hemiparesis or speech disturbances; recovery is often complete, although persistent deficits have been reported [52]. PRES, often occurring in early therapy, is linked to disturbed cerebrovascular autoregulation, with variable symptoms such as seizures, visual disturbances, and altered mental status; arterial hypertension, chemotherapy, and corticosteroids are among the main responsible. Moreover, PRES particularly manifests after almost 4 weeks of cranial radiotherapy [47].

Corticosteroid-induced psychosis has also been documented, though treatment guidelines are lacking; supportive care may include hypnotics, tranquilizers, or antipsychotics like risperidone in severe cases [53,54].

Peripheral neuropathy—primarily caused by vincristine—is common and typically reversible; however, recovery may take months [55].

### 5.4. Endocrinopathies

Endocrine complications are relatively common during childhood ALL therapy. Indeed, a reduction in growth hormone levels and, consequently, growth retardation are observed; however, the majority of children—particularly those not receiving radiotherapy—experience sufficient growth post-therapy, though with a trend toward a slightly reduced final height [56]. Significant weight gain affects up to 40% of patients, largely due to corticosteroid exposure and reduced physical activity, which can induce insulin resistance, hyperglycemia, and prediabetes—conditions that may necessitate dietary adjustments or insulin therapy [57,58]. Importantly, both obesity and hyperglycemia have been associated with worse event-free survival (EFS) [59,60]. Corticosteroids also suppress the hypothalamic–pituitary–adrenal (HPA) axis, leading to secondary adrenal insufficiency in nearly all patients. This effect may persist for months and may be worsened by fluconazole co-administration [61]. The duration of adrenal suppression has been associated with glucocorticoid receptor (GR) gene variants [62]. During ALL therapy, high levels of triglycerides and cholesterol are common and are primarily linked to corticosteroid and asparaginase use [63,64,65]. Some studies suggest potential links between hypertriglyceridemia and complications such as ON and thrombosis [64,66].

### 5.5. Hepatotoxicity

Hepatotoxicity is a well-recognized adverse effect associated with the treatment of childhood ALL. It typically occurs during maintenance therapy and is mainly caused by the metabolites of 6-MP and MTX [67]. At low doses, MTX toxicity mainly results in moderate myelosuppression and liver damage, whereas HD-MTX—typically a 24 h intravenous infusion of 5 g/m^2^ followed by leucovorin rescue—is linked to acute and severe kidney, neurological, and hepatic toxicities [68].

Thiopurines commonly induce liver toxicity, typically presenting as elevated aminotransferases without other liver dysfunction signs. This effect is linked to high levels of methylmercaptopurine metabolites (MeMPs), though the precise mechanism remains unclear [68].

SOS is a severe, potentially life-threatening liver toxicity—mainly associated with 6-thioguanine (6-TG)—caused by impaired hepatic microcirculation. While SOS is common and dangerous after stem cell transplantation, during chemotherapy, it occurs less frequently and is usually manageable. In clinical trials, 10–25% of patients receiving 6-TG developed SOS or thrombocytopenia, and around 2.5% showed chronic liver damage such as nodular regenerative hyperplasia (NRH) [68,69].

Both SOS and NRH often coincide with thrombocytopenia, though high DNA-TG levels are not associated with greater SOS or thrombocytopenia risk. In the TEAM pilot study, no serious hepatic adverse events were observed, suggesting that the TEAM strategy does not increase SOS/NRH risk [70]. Nonetheless, when combining thiopurines with newer agents like inotuzumab—which can also trigger SOS, particularly before stem cell transplantation—careful monitoring and further research are recommended [71].

### 5.6. Thromboembolism

TE is the most frequent thrombotic event in pediatric ALL, with approximately half of cases involving the CNS [66]. The incidence of symptomatic venous TE ranges between 2% and 8%, although asymptomatic events occur in up to 70% of patients [72]. Identified risk factors for TE include the underlying leukemia, older age, the presence of central venous catheters, prolonged immobilization, concurrent infections, systemic inflammation, and treatment with asparaginase and/or corticosteroids [72]. Conversely, inherited thrombophilia and common germline polymorphisms do not appear to play a significant role, or their contribution remains uncertain [73,74]. Among the venous thrombotic events, cerebral vein thromboses are associated with the highest fatality rates in children. Despite the clinical burden, the role of thromboprophylaxis in this population remains to be clearly defined. Further studies have assessed the efficacy and safety of prophylactic anticoagulation, particularly using novel oral anticoagulants to reduce the incidence and severity of thrombotic complications during ALL therapy [73,74,75].

### 5.7. High-Dose Methotrexate-Induced Nephrotoxicity

Alkalinization combined with aggressive hydration significantly reduces the risk of HD-MTX-induced nephrotoxicity [76]. However, some patients could experience severe renal impairment, which may in turn hinder MTX clearance [76,77]. This nephrotoxicity is primarily caused by intrarenal precipitation of MTX crystals, often related to suboptimal hydration and urine alkalization [78]. Typically, plasma creatinine levels peak within a few days of HD-MTX administration and return to baseline within a few weeks. Notably, most patients are able to tolerate subsequent full-dose HD-MTX courses without recurrent renal toxicity [76].

To mitigate life-threatening complications such as myelosuppression and mucositis, higher doses of folinic acid rescue are critical. Nevertheless, whether excessive leucovorin rescue may inadvertently increase relapse risk remains an unresolved concern [79,80].

Several external agents, including proton pump inhibitors, non-steroidal anti-inflammatory drugs, and certain dietary elements, are potential modifiers of MTX pharmacokinetics [37,81,82].

### 5.8. Asparaginase-Associated Hypersensitivity, Allergy, and Pancreatitis

Asparaginase therapy causes several toxicities primarily due to the systemic depletion of asparagine and the consequent disruption of protein synthesis. These toxicities can affect approximately 20–25% of patients and may necessitate treatment discontinuation, which increases the risk of relapse—particularly in the CNS [37].

Asparaginase can trigger the formation of neutralizing antibodies, either with clinical signs of hypersensitivity or without symptoms (silent inactivation). Detection of silent inactivation requires therapeutic drug monitoring of plasma asparaginase activity [83,84,85]. Allergic reactions are highly variable in incidence, ranging from 3% to 75%, depending on the type, dosage, administration route, and duration of therapy [85,86]. Reactions typically occur after the first or second dose and are almost universally associated with undetectable asparaginase activity. Clinical manifestations range from mild cutaneous symptoms to severe systemic responses, including urticaria, bronchospasm, angioedema, and hypotension. Premedication with corticosteroids and antihistamines, along with prolonged infusion times, may attenuate symptoms but does not prevent inactivation [37].

Asparaginase-associated pancreatitis (AAP) occurs in 2–18% of patients, depending on cumulative exposure and monitoring intensity [37]. The condition typically manifests within two weeks of asparaginase administration (median: 11 days for PEG-asparaginase). Severe AAP is defined by imaging evidence of necrosis/hemorrhage or persistent symptoms and enzyme elevation for over 72 h [37].

## 6. Targeted Therapy

In recent decades, a wide range of novel biomarkers and genetic alterations have been identified in ALL, which has aided the development of precision therapies targeting cells based on the modification expressed. Nowadays, the identification of genetic modifications and their significance is pivotal in developing improved opportunities and tailored therapies.

The latter are classified into two major groups: small molecules and monoclonal antibodies. Targeted therapies are tailored to specifically target the cancer cells of an individual, unlike chemotherapy, and they are integrated into induction protocols to improve the rate of success in specific genetic and molecular subgroups; thus, depending on the different alterations identified in the patient, the therapy will be tailored to their needs [87].

Currently, childhood ALL can be divided into several genetic subgroups, each associated with different and specific patient prognoses, whose mutations or chromosome translocations lead to the misregulation of numerous metabolic pathways commonly associated with increased activity and cell survival. Consequently, new methods for targeting and inhibiting the overexpression of these pathways have been proposed and are nowadays being implemented, leading to an improved prognosis and a decrease in treatment toxicity (Figure 3 and Figure 4). Nevertheless, even though different targeted immunotherapies have revolutionized the treatment of B-ALL, progress in T-ALL has been considerably slow due to the lack of well-defined surface antigens that can be safely targeted without affecting normal T cells. Among the emerging therapeutic targets, CD38 has attracted growing interest in T-ALL. It is physiologically expressed on thymocytes and activated T cells, while its expression in normal hematopoietic and nonhematopoietic tissues remains limited. Importantly, CD38 is aberrantly expressed in several hematologic malignancies, including subsets of T-ALL, making it a potential candidate for targeted therapy [88].

### 6.1. Tyrosine Kinase Inhibitors

The presence of the Philadelphia chromosome, which results in the production of a BCR-ABL1 fusion protein with constitutive protein kinase activity, is a poor prognostic marker. Similar to BCR-ABL1-positive ALL, patients with BCR-ABL-like ALL have a reduced five-year disease-free survival (DFS), and they are caused by the different mutations previously described, which similarly result in the overexpression and altered modulation of common surface receptor/kinase signaling pathways. Therefore, one of the major groups of therapeutic molecules employed against Ph+ and Ph like+ ALL is TKIs.

TKIs are highly selective and offer convenience due to their oral administration, and their role is to reduce tyrosine kinase phosphorylation, thus inhibiting proliferation, cell division in the G1 phase, angiogenesis, and the overall survival (OS) of cancer cells [89].

Consequently, the combined use of TKIs with chemotherapy has become the standard of care in the treatment of patients with Ph-like ALL and ABL-class gene fusions.

TKIs can be classified as type I or type II inhibitors, based on the strategy by which inhibition is achieved, namely, competitive inhibition or allosteric inhibition, which depends on whether they recognize an active/phosphorylated or inactive/unphosphorylated kinase domain, respectively. Although inhibitors have proven to be an option, they show scarce binding selectivity, inhibiting other kinases, thereby causing off-target side effects leading to cardiac, pulmonary, gastrointestinal, and endocrine toxicity issues, especially in children [90].

Additionally, TKIs can also be divided into generations depending on when they were introduced for standard treatment. The first trial using imatinib, a first-generation TKI, took place in 2004 and was followed by the establishment of the use of the drug as an addition to the standard protocol of chemotherapy, leading to the avoidance of bone marrow transplant while maintaining a 5-year OS of 70% [91,92].

Nevertheless, the following efforts were focused on overcoming the challenge posed by a subgroup of patients who developed intolerance or resistance to imatinib, which led to the introduction of novel-generation TKIs for treatment such as dasatinib and nilotinib. The cause of intolerance or imatinib resistance was highlighted by the presence of the point mutation ABL1 T315I, known as a “gatekeeper”, which causes the loss of hydrogen bonds, which are critical for drug binding. New-generation TKIs are able to bypass the mutation and exhibit potent activity in both wild-type and mutant BCR-ABL1 ALL. The second-generation TKI dasatinib differs from its precursor in several ways; in particular, it has greater inhibitory potency due to its ability to bind both activated and non-activated conformations of the kinases as a type II competitive inhibitor, in opposition to imatinib, whose target is only the activated isoform. Additionally, dasatinib is able to modulate other specific anti-leukemic pathways involving MAPK and BCL2, targeting other non-BCR/ABL kinases such as the SRC family and blocking STAT-5 downstream pathways. Moreover, dasatinib has demonstrated the ability to cross the blood–brain barrier, thus being able to carry out its function in the CNS and significantly improve EFS and OS compared to imatinib treatment [91].

On the other hand, although the third-generation TKI ponatinib is one of the newest treatments and has demonstrated potent activity in both wild-type BCR/ABL1-positive ALL and resistant variants, it is also associated with an increased risk of thrombosis and pancreatitis in children [93].

Among the other kinases that may be overexpressed in pediatric ALL and could potentially be a target for therapies, FMS-like tyrosine kinase 3 (FTL3) and tropomyosin receptor kinase (TRK) are currently under study. Furthermore, clinical trials following FTL3 and TRK inhibitors, in addition to the standard protocol of chemotherapy, have started in pediatric ALL patients. In particular, the FTL3 inhibitor lestaurtinib and the TRK inhibitor, larotrectinib have shown great promise [94,95].

### 6.2. JAK-STAT Inhibitors

Other mutations with prognostic function are *CRLF2* rearrangements, which lead to the activation of PI3K/AKT/mTOR and JAK–STAT signaling; both of these can be modulated as targets and control several functions in the cell. In particular, the interleukin-7 (IL-7)/Janus kinase (JAK)/signal transducer and activator of transcription (STAT) signaling pathway is involved in both T- and B-cell development and cell survival, proliferation, and apoptosis by modulating the synthesis of the anti-apoptotic BCL2 protein. Therefore, an alteration in the modulation of the JAK/STAT pathway, such as overactivation, as established in ALL patients, leads to variations in several fundamental processes.

Concurrently, clinical trials regarding the employment of ruxolitinib as a JAK-STAT inhibitor together with chemotherapy as treatment for ALL patients are ongoing. Although JAK inhibitors show promise, *CRLF2*-rearranged Ph-like ALL patients respond poorly to single-agent therapy, prompting the identification of combination regimens via high-throughput drug screening. Ruxolitinib improved the efficacy of standard-of-care drugs employed in ALL treatment and increased induction rates in synergy with vincristine, dexamethasone, and l-asparaginase in most *CRLF2*-rearranged Ph-like ALL models. Nevertheless, ruxolitinib has also been proven to be effective in Ph-like ALL, a high-risk molecular subtype that has a gene expression profile similar to that of Ph+ ALL but does not harbor the *BCR-ABL1* gene fusion. Despite the therapeutic properties of the drug, severe adverse effects have been reported for the use of ruxolitinib in pediatric ALL, such as thrombocytopenia, neutropenia, and anemia. Other JAK-STAT inhibitors are currently under investigation, such as tofacitinib and peficitinib, which are always employed and integrated into the chemotherapy backbone; in particular, NCT06128070 is a trial that highlights the potential of ruxolitinib together with tacrolimus and MTX for the prevention of graft versus host disease in pediatric and young adult patients undergoing allogeneic hematopoietic cell transplant for not only ALL but also acute myeloid leukemia and myelodysplastic syndrome [96].

Moreover, the IL-7R/JAK/STAT signaling pathway also plays a crucial role in T-cell development and homeostasis, primarily by promoting cell survival. Comprehensive genomic analyses of T-ALL have identified activating mutations in *IL7R*, *JAK1*, *JAK3*, and/or STAT5 in approximately 20–30% of cases, with a higher frequency observed in ETP-ALL subtypes [97]. Pharmacological inhibition of JAK1/2 with ruxolitinib has shown antileukemic efficacy in primary xenograft models of ETP-ALL, both in the presence and absence of *JAK*–*STAT* mutations, and it has been reported to counteract IL-7-induced hyperactivation [98,99]. In addition, tofacitinib has demonstrated preclinical activity against T-ALL cells harboring *IL7R* or *JAK1*/*JAK3* mutations [100].

### 6.3. mTOR Inhibitors

On the other hand, PI3K/AKT/mTOR signaling is also involved in several important functions such as cell death, metabolism, and proliferation, and an abnormal activation is seen in ALL patients. Moreover, dysregulation of these pathways has been associated with resistance to chemotherapy [87]. mTOR inhibitors, such as idelalisib, everolimus, and temsirolimus have been shown to inhibit cell growth and revert glucocorticoid resistance, working synergistically with other chemotherapeutic agents [101].

Furthermore, aberrant activation of the PI3K/Akt/mTOR signaling pathway in T-ALL commonly results from loss-of-function mutations or deletions of the tumor suppressor gene *PTEN*, detected in approximately 10% of cases. In addition, gain-of-function mutations affecting the regulatory or catalytic subunits of PI3K have been reported [99,100]. This dysregulation enhances cell growth, metabolism, and proliferation while reducing apoptosis, and it has been associated with glucocorticoid resistance [100,102].

Several clinical trials are currently evaluating mTOR inhibitors such as everolimus and temsirolimus [102]. In a phase I/II trial (NCT00968253), everolimus combined with HyperCVAD chemotherapy in relapsed or refractory (R/R) ALL demonstrated moderate efficacy and good tolerability [102]. Another phase I study (NCT01523977) confirmed the feasibility of the use of everolimus alongside prednisone, vincristine, PEG-asparaginase, and doxorubicin in relapsed ALL [103]. An ongoing phase I trial (NCT03740334) is currently assessing the combination of ribociclib, everolimus, and dexamethasone in patients aged 1–30 years with relapsed ALL [104].

### 6.4. BCL-2 Inhibitors

Other fundamental proteins whose role in the intrinsic mitochondrial apoptosis pathway is essential are those belonging to the BCL-2 protein family. The latter’s overexpression contributes to apoptosis resistance in leukemic cells, and venetoclax, a BCL-2 inhibitor, is able to block the anti-apoptotic regulator’s expression, modulating the cell death pathway. Both venetoclax and navitoclax act as BH3 mimetics and are selective inhibitors of BCL-2; the latter also selectively inhibits BCL-XL, which displaces the pro-apoptotic proteins BIM and BAX, leading to the permeabilization of the mitochondrial outer membrane, cytochrome c release, and the intracellular activation of caspases, contributing to apoptosis. BCL-2 overexpression is associated with glucocorticoid resistance in ALL cells; however, both venetoclax and navitoclax have been shown to mitigate it [105].

Additionally, clinical trials regarding venetoclax in combination with chemotherapy in pediatric patients with R/R ALL are ongoing and are showing promise [106].

Considering these findings, the anti-apoptotic protein BCL-2 has become a major focus of targeted therapy development in ALL. Several BH3 mimetics promoting apoptosis through the inhibition of BCL-2 family proteins have been evaluated [107,108]. BH3 profiling studies revealed that the dependency on BCL-2 or BCL-XL in T-ALL correlates with the maturation stage of the malignancy: while most T-ALL cases are BCL-XL-dependent, ETP-ALL exhibits preferential BCL-2 dependence and selective sensitivity to venetoclax (ABT-199) [109]. Earlier agents such as ABT-737 and its oral analog ABT-263 (navitoclax) were limited by thrombocytopenia due to BCL-XL inhibition in megakaryocytes, a drawback that was overcome by the development of the BCL-2-selective inhibitor venetoclax [107,108]. Approved by the FDA for CLL and AML in elderly patients [107,110], venetoclax has demonstrated potent antileukemic activity in T-ALL. Despite the potential for resistance as a single agent [111], combination regimens have improved chemosensitivity, reduced toxicity, and mitigated resistance mechanisms. Preclinical and clinical data show significant activity of venetoclax in combination with chemotherapy, decitabine, nelarabine, or bortezomib (VEBO regimen), particularly in R/R T-ALL [111,112,113]. Ongoing phase I trials (NCT03236857 and NCT03181126) are further evaluating its safety, pharmacokinetics, and efficacy in pediatric and young adult patients with R/R ALL [104].

### 6.5. Menin Inhibitors

Another mutation associated with poor prognosis in ALL is the *KMT2A* rearrangement, which leads to the formation of a unique multi-protein fusion complex. Menin interacts with the fusion proteins, stabilizing them, binding to chromatin and DNA, and enabling the dysregulation of gene expression, in particular leading to the overexpression of genes such as *HOX* and *MEIS1*, eventually reaching leukemic transformation [114]. Therefore, this complex has been identified as a therapeutic target through menin inhibition. In particular, revumenib is currently under study as a menin inhibitor capable of blocking menin from combining with the KMT2A protein [115].

### 6.6. Proteasome Inhibitors

Another class of molecules that have been used as targeted therapies to regulate protein homeostasis are proteasome inhibitors, among which the most studied are bortezomib, ixazomib, and carfilzomib. Targeting proteasomal degradation pathways has shown benefit in pediatric T-ALL; in particular, carfilzomib, as a second-generation agent, is a highly selective proteasome inhibitor that shows reduced peripheral neurotoxicity in comparison to bortezomib. A phase II study regarding the use of carfilzomib in combination with induction chemotherapy in children with R/R ALL highlighted the promising results of carfilzomib in combination with VXLD chemotherapy, with an overall remission rate of 67% at the end of consolidation for pediatric patients with highly advanced ALL [116].

### 6.7. MEK Inhibitors

Another class of molecules that are currently under investigation as targeted therapies for pediatric ALL is MEK inhibitors, whose clinical trials are progressing and show promise for their future clinical application. In particular, MEK kinase is a component of the RAS-RAF-MEK-ERK pathway, and approximately 40% of R/R ALL patients carry different RAS pathway mutations. Selumetinib acts as an MEK inhibitor and has been shown to work synergistically with dexamethasone in RAS pathway mutations associated with pediatric ALL in preclinical studies. Dexamethasone-mediated blast apoptosis is induced by increasing the pro-apoptotic protein Bim, whose inactivation is caused by phosphorylation by ERK. Therefore, by preventing the phosphorylation of MEK, ERK is inhibited, hence increasing apoptosis mediated by Bim. Currently, a phase I/II clinical trial is assessing the efficacy of dexamethasone and selumetinib synergy in pediatric and adult R/R ALL patients [117].

### 6.8. CDK Inhibitors

Furthermore, another pathway commonly disrupted in cancer is the CDK 4/6-p16-retinoblastoma (Rb) pathway, whose overactivation leads to abnormal cell proliferation and survival. In particular, mutations in proteins related to pathways such as NOTCH lead to the upregulation of cyclin D3, resulting in CDK4/6 hyperactivation, the phosphorylation of Rb, and eventually cell cycle progression. Therefore, inhibitors of CDKs, particularly selective inhibitors of CDK4/6 such as ribociclib and palbociclib, are currently under study. A phase 1 trial evaluated palbociclib in combination with standard four-drug re-induction chemotherapy in children and young adults with R/R B- and T-ALL and lymphoma [118]. Furthermore, another phase I trial investigated ribociclib in synergy with everolimus, an mTOR inhibitor, and dexamethasone in R/R pediatric ALL [119].

### 6.9. NOTCH Signaling Inhibition

Mutations in the NOTCH signaling pathway are common in T-ALL, prompting the development of targeted therapeutic strategies such as γ-secretase inhibitors (GSIs), soluble NOTCH proteins, and inhibitory peptides [104,120]. GSIs are small molecules that block all four NOTCH receptors, thereby suppressing NOTCH signaling and inducing cell cycle arrest (G0–G1) and apoptosis in specific T-ALL cell lines [121]. However, early clinical trials yielded limited results, mainly due to gastrointestinal toxicity—notably grade 3/4 diarrhea resulting from NOTCH1 and NOTCH2 inhibition in the intestine, as reported in the Dana-Farber phase I trial of MK-0752 [122].

To overcome these limitations, current research focuses on combination therapies that enhance antileukemic efficacy while reducing toxicity and GSI resistance [121]. For example, PF-03084014 (a reversible, non-competitive GSI developed by Pfizer) has shown that intermittent dosing may mitigate systemic toxicity, which appears to be time- and dose-dependent [121,122]. Preclinical studies combining PF-03084014 with glucocorticoids demonstrated synergistic antileukemic effects and reduced intestinal damage due to glucocorticoid-mediated protection [122,123]. Additional in vitro studies have reported synergism between GSIs and compounds such as withaferin A, rapamycin, vorinostat, or chloroquine, with the latter also associated with fewer adverse effects [124,125].

Overall, GSI monotherapy appears insufficient in *NOTCH1*-mutated T-ALL, while combination regimens represent a promising approach to improve therapeutic outcomes and minimize gastrointestinal toxicity [126].

Beyond the limitations and ongoing optimization of GSI-based therapy, several alternative strategies targeting the NOTCH signaling pathway have been explored. These include ADAM10 inhibitors and SERCA inhibitors such as CAD204520, which interfere with NOTCH receptor processing [104]. Another promising approach involves the Mastermind-inhibiting peptide SAHM1, an α-helical molecule currently under active investigation. Moreover, monoclonal antibodies directed against the NOTCH1 receptor have demonstrated preclinical efficacy. Notably, OMP-52M51, a monoclonal antibody generated by immunizing mice with human NOTCH1 protein fragments, exhibited antitumor activity both in vitro and in vivo in xenograft T-ALL models [126].

## 7. Immunotherapy

### 7.1. Blinatumomab

Blinatumomab is a 55 KDa bispecific T-cell engager (BiTE^®^) antibody construct recommended for the treatment of ALL [127,128]. It is composed of two single-chain variable fragments (scFvs) derived from murine monoclonal antibodies, one binding to CD3 on T cells and the other binding to CD19 on B cells, connected by a flexible non-immunogenic glycine-serine linker. In particular, CD19 is expressed on the surface of precursor B cells and is involved in the self-renewal of leukemic cells, while CD3 is a TCR responsible for T-cell activation [127,129,130,131] (Figure 5). After blinatumomab administration, T cells establish an immune synapse with CD19+ target cells, releasing cytolytic granules containing perforin and granzymes, and producing inflammatory cytokines such as interferon-gamma (IFN-γ), tumor necrosis factor-alpha, and interleukins (IL-2, IL-6, and IL-10) [127,130,132,133].

Blinatumomab induces a rapid redistribution of T cells from the bloodstream within 2–6 h of infusion, returning to baseline within 7–10 days, suggesting migration rather than depletion [128,134,135]. This early T-cell disappearance is likely due to adhesion to the endothelium or extravasation upon CD3 engagement [128,135]. Memory effector T cells (CD45RA^−^/CCR7^−^) predominantly expand, particularly CD8^+^ subsets, under cytokine stimulation [128,134,136]. Activation markers (CD69 and CD25), adhesion molecules (CD2 and LFA-1), and the endothelial activation marker angiotensin II rise transiently post-infusion [128,137].

B-cell depletion is rapid and profound, with CD19^+^ cells being undetectable within 48 h due to apoptosis, as indicated by annexin V [128,135,136,138]. In non-responders, B cells may persist [31]. T- and B-cell engagement triggers a surge in pro-inflammatory cytokines (IL-10, IL-6, and IFN-γ), peaking on day 1 and resolving by day 2. These responses are not predictive of treatment efficacy [128,138].

Blinatumomab pharmacokinetics is characterized by a short half-life (≈1–2 h), a low volume of distribution, and minimal renal clearance, and it requires continuous intravenous infusion [128,139]. It exhibits potent cytotoxicity at low concentrations due to high CD19 affinity and T-cell lytic potential [128,135].

In pediatric B-ALL, blinatumomab is administered via continuous intravenous infusion for 28 days per cycle [12]. In children, its efficacy and safety were first demonstrated in a phase I/II study (NCT01471782) and the RIALTO phase II trial (NCT02187354), which showed encouraging response rates both in overt leukemia and in MRD-positive patients, including those relapsing after allogeneic hematopoietic stem cell transplantation (allo-HSCT) [5,140]. These studies confirmed high rates of complete remission with minimal toxicity, primarily consisting of manageable neurological adverse events and cytokine release syndrome (CRS). Real-world data from the NEUF expanded-access program corroborated these findings, with more than 50% of heavily pretreated children achieving remission, and excellent responses in MRD-positive settings [141]. The role of blinatumomab has been further validated in pivotal randomized phase III trials: in AALL1331 (COG, NCT02101853) for children and adolescents with low-, intermediate-, or high-risk first relapses and in the International B-ALL Consortium study (NCT02393859) for high-risk relapsed disease.

Both trials showed significant improvements in EFS, DFS, MRD clearance, and OS, establishing blinatumomab as a new standard of care in post-reinduction consolidation therapy for relapsed B-ALL across different risk categories [142,143]. However, in the AALL1731 trial, patients with isolated extramedullary relapse had poor outcomes regardless of whether they received chemotherapy or blinatumomab, underscoring the need for alternative approaches in this subset. Blinatumomab has also been included in frontline treatment regimens. A pilot study (EudraCT 2016-004674-17) tested its use in infants with KMT2A-rearranged B-ALL by adding one blinatumomab cycle post-induction to the Interfant-06 protocol. The results were compelling: MRD-negative rates significantly increased, and 2-year DFS and OS improved markedly (81.6% and 93.3%, respectively) compared to historical controls. This led to its inclusion in the ongoing Interfant-21 trial (NCT05327894) [144,145]. The AIEOP-BFM ALL 2017 trial (NCT03643276) became the first randomized phase III study to directly compare chemotherapy with and without blinatumomab in intermediate-risk patients, and to test two cycles of blinatumomab instead of intensive chemotherapy blocks in high-risk pediatric B-ALL. Preliminary data indicated a favorable toxicity profile in the blinatumomab arms, with most adverse events being non-severe neurologic events [146]. Similarly, in standard-risk B-ALL, the COG phase III trial (NCT03914625) demonstrated superior 3-year DFS in both average- and high-risk groups treated with blinatumomab (97.5% vs. 90.2% and 94.1% vs. 84.8%, respectively). These results prompted early closure of randomization and firmly positioned blinatumomab as a new standard of care for post-induction consolidation in standard-risk pediatric B-ALL. Notably, an increased rate of catheter-related infections was observed during and after blinatumomab administration, warranting further investigation [147,148]. Beyond standard protocols, blinatumomab has proven valuable in chemotherapy-intolerant patients, as shown by the UK ALL group in a cohort of 105 children. In these cases, 1–2 cycles of blinatumomab used as a consolidation replacement led to 97% MRD-negative CRs, with survival outcomes comparable to patients receiving conventional chemotherapy. Based on these data—including supportive results in adult clinical trials—the FDA has approved blinatumomab for use as consolidation therapy in patients aged ≥28 days with newly diagnosed B-ALL, and EMA approval is expected. Ongoing frontline trials are now incorporating blinatumomab in various high-risk subgroups: EsPhALL2022/COG AALL2131 (NCT06124157) for Ph+ and Ph-like B-ALL and early-relapse protocols where blinatumomab may replace intensive chemotherapy during early consolidation [149,150,151].

Finally, a subcutaneous formulation of blinatumomab, tested in adults with r/r B-ALL and showing high MRD clearance rates, is under consideration for pediatric trials, potentially reducing the logistical burden of continuous infusion [152].

In summary, blinatumomab is increasingly integrated across all stages of pediatric B-ALL therapy, from relapse management to upfront treatment, offering a highly effective and better-tolerated alternative to intensive chemotherapy—especially valuable in improving MRD responses, reducing toxicities, and possibly minimizing the need for stem cell transplantation.

### 7.2. Daratumumab

Daratumumab is a human immunoglobulin G1κ monoclonal antibody that binds to CD38, and it has been demonstrated to be safe and effective in patients with refractory multiple myeloma, as approved by the FDA. Nowadays, preclinical trials show that CD38 is highly expressed at both diagnosis and relapse steps in many pediatric hematological malignancies, with a strong expression in ALL; therefore, novel studies are highlighting the therapeutic role of CD38-targeting monoclonal antibodies, such as daratumumab and isatuximab, in pediatric hematological malignancies like ALL [153].

CD38 is a type II transmembrane glycoprotein that has been implicated in the regulation of cytoplasmic calcium flux and that mediates signal transduction in immune cells, and it is vastly found as a surface marker of immune cells, such as terminally differentiated B, and activated T cells, but in the majority of cases, it is more commonly found on plasma B cells [88].

CD38 is a multifunctional cell surface protein endowed with receptor/enzymatic functions, such as the regulation of extra-cellular nucleotide homeostasis, intra-cellular calcium fluxes, and cell adhesion, and it cooperates in signal transduction acting in synergy with CD31 (PECAM-1). Therefore, CD38 is also considered a modulator of immune cell activation, and an analysis of CD38 allows for an assessment of the inflammatory status of the patient [153].

Daratumumab induces lysis through cytotoxic mechanisms by aiding the immune modulatory response of the patient, leading to the apoptosis of cancer cells (antibody- and complement-dependent phagocytosis). In particular, a phase I–II trial found that daratumumab, combined with backbone chemotherapy, may effectively serve as a bridge to hematopoietic stem cell transplantation (HSCT) in children and young adults with R/R T-cell ALL/LL [154].

It has been demonstrated that blasts from pediatric and young adult patients with newly diagnosed T-ALL express CD38 on their surfaces and that this expression remains stable after one month of multiagent chemotherapy. The antileukemic activity of daratumumab has been confirmed in patient-derived xenograft (PDX) models, supporting its potential role in T-ALL treatment. Moreover, a phase II clinical trial is currently investigating the combination of daratumumab with standard chemotherapy in children and young adults with R/R T- or B- ALL [104].

Although most pharmacokinetic data on daratumumab are derived from studies on multiple myeloma, these findings provide valuable insights into its pharmacokinetic behavior, which is expected to be similar in acute leukemias due to comparable CD38 expression and target-mediated clearance. Early clinical trials, including the GEN501 and SIRIUS studies, demonstrated that intravenous daratumumab exhibits typical IgG monoclonal antibody pharmacokinetics, characterized by a bi-exponential decline in serum concentration after infusion [155]. Maximum serum concentrations increased approximately dose-proportionally from 1 to 24 mg/kg after the first full dose and greater than dose-proportionally after multiple doses [155]. Estimated clearance decreased with increasing dose levels and over time, reflecting target-mediated drug disposition, while the volume of distribution remained relatively small, indicating confinement to the vascular compartment [155,156]. Evidence of drug accumulation was observed with repeated weekly dosing, with accumulation factors ranging from 1.8 to 3.5 [157]. These pharmacokinetic features are consistent with those of monoclonal antibodies and contribute to the sustained therapeutic activity of daratumumab when adequate dosing schedules are maintained [155]. Daratumumab has shown a favorable and manageable safety profile in both monotherapy and combination settings [155]. In the GEN501 and SIRIUS studies, no maximum tolerated dose was reached up to 24 mg/kg, and no patient discontinued treatment due to drug-related adverse events [158]. The most frequently reported events included fatigue, nausea, anemia, back pain, cough, upper respiratory tract infections, thrombocytopenia, and neutropenia, with no clear dose-related pattern [158]. Importantly, no correlation was found between serum daratumumab concentrations and hematologic toxicity or infection rates [159]. Infusion-related reactions (IRRs), mostly occurring during the first administration, are the most characteristic adverse events, generally being mild to moderate and presenting with respiratory or allergic symptoms (e.g., nasal congestion, cough, chills, and dyspnea) [155]. These are effectively managed with corticosteroids, antihistamines, and acetaminophen. To improve tolerability, a split first-dose regimen (8 mg/kg on two consecutive days) was introduced, which significantly reduced infusion time without affecting pharmacokinetic exposure or increasing adverse reactions [155].

Similar to other monoclonal antibodies, the major adverse effects that have been reported are infusion reactions, in addition to overall fatigue and upper respiratory infections [160].

### 7.3. Inotuzumab Ozogamicin

Inotuzumab ozogamicin (InO) is a humanized anti-CD22 monoclonal antibody conjugated to the cytotoxic agent calicheamicin [161,162,163,164]. Its efficacy was first demonstrated in adult patients with R/R B-ALL in the phase III INO-VATE trial (INotuzumab Ozogamicin trial to inVestigAte Tolerability and Efficacy), which showed significantly higher remission rates and improved survival compared to standard-of-care chemotherapy [161,165,166,167]. More recently, it has shown increasing promise in the treatment of pediatric B-cell ALL (pB-ALL), particularly in the R/R setting [162,168]. On 6 March 2024, the U.S. Food and Drug Administration granted approval for the use of InO (Besponsa, Pfizer) in pediatric patients aged 1 year or older with R/R CD22-positive BCP-ALL [164]. CD22 is a B-cell-specific transmembrane protein belonging to the siglec and immunoglobulin superfamilies, expressed in over 90% of B-lineage ALL cases [164,169]. It regulates B-cell receptor (BCR) signaling through the cis- and trans-binding of ligands containing sialic acid, influencing activation and migration [164,169,170]. Critically, CD22 undergoes rapid internalization upon ligand binding, a property that enables its use for antibody–drug conjugates (ADCs) like InO (13). InO consists of a monoclonal antibody targeting CD22, a linker, and a cytotoxic agent called calicheamicin, which is released intracellularly following endocytosis, leading to DNA strand breaks and apoptosis [164,171]. In vitro studies have shown that internalization efficiency affects cytotoxicity, while clinical data indicate that a higher CD22 surface density is correlated with better clinical outcomes [164]. Patients with normal cytogenetics and elevated CD22 expression derive the most benefit from inotuzumab therapy [164]. Conversely, cases with *KMT2A* rearrangements, a high-risk cytogenetic subtype, are often associated with lower CD22 expression and persistent MRD following treatment. Notably, CD22 expression can decrease at relapse after inotuzumab exposure, although the underlying mechanisms remain unclear [164,172]. The pivotal phase I/II ITCC-059/AALL1621 study (NCT02981628), conducted by the Children’s Oncology Group and the Innovative Therapies for Children with Cancer consortium, assessed the safety and efficacy of inotuzumab in pediatric patients aged 1 to 21 years with R/R B-ALL [173,174]. The study identified a recommended phase 2 dose using a weekly fractionated schedule and showed promising results, with complete remission achieved in about 80% of patients and many reaching MRD negativity after just one or two cycles [173]. InO has also proven valuable as a bridge to HSCT and CAR T-cell therapy in some cases, particularly when rapid disease control is required prior to cell collection or infusion [175]. However, the use of InO in this context is tempered by a notable risk of hepatic veno-occlusive disease (VOD), particularly in patients undergoing HSCT shortly after InO administration [176].

Regarding cardiac safety, a population exposure–response analysis demonstrated that InO has no clinically meaningful effect on QT interval prolongation. The expected median change in QTcF and QTcS intervals at both therapeutic and supratherapeutic concentrations was <5 ms, with an upper 97.5th percentile of <10 ms [177]. In the INO-VATE trial, QT prolongation was reported in only 1% of patients (grade 2), with no cases of QTcF >500 ms, torsade de pointes, or sudden death observed. Nevertheless, due to the potential influence of comorbidities and concomitant medications, electrocardiogram (ECG) and electrolyte monitoring are recommended before and during treatment, particularly in patients at risk of QT prolongation [177].

InO is currently being studied as a potential alternative to chemotherapy in patients with first-relapse BCP-ALL. Ongoing clinical trials are either evaluating InO as a single-agent reinduction therapy, especially in high-risk cases, or comparing it directly to standard chemotherapy regimens [178].

## 8. CAR-T

Over the past decade, immunotherapy using endogenous T cells has represented a promising strategy for treating R/R ALL, offering an alternative to overcome chemotherapy resistance. While conventional multiagent chemotherapy has dramatically improved survival in pediatric B-ALL, with current 5-year OS rates exceeding 90% in high-income countries, outcomes remain poor in relapsed cases, where 5-year OS drops to 19–52% [179,180].

Immune effector cell therapies, particularly chimeric antigen receptor (CAR) T-cell therapy, have significantly improved outcomes in patients with R/R leukemia. CAR T cells are genetically modified polyclonal T or natural killer (NK) cells that express fusion proteins, which target specific surface molecules on tumor cells [181]. CAR T-cell therapy is an advanced form of adoptive cell therapy (ACT) developed to overcome the limitations of conventional T-cell responses, including HLA restriction and insufficient TCR diversity. CAR-T cells are genetically engineered to express synthetic receptors that recognize tumor-associated antigens in an MHC-independent manner, enhancing tumor targeting. Structurally, CARs are composed of an extracellular antigen-binding domain (typically a single-chain variable fragment, scFv), a hinge region, a transmembrane domain, and intracellular signaling domains. The evolution of CAR design has led to multiple generations, with the second generation incorporating co-stimulatory domains to improve T-cell proliferation, persistence, and antitumor activity (Figure 6) [182].

CAR T-cell therapy has received FDA approval for R/R B-ALL and diffuse large-B-cell lymphoma (DLBCL) [179]. However, extending this success to T-cell malignancies remains challenging due to the lack of selective target antigens distinguishing normal T cells from malignant T cells. Shared antigen expression leads to CAR-T cell fratricide, where modified T cells attack each other, and to profound T-cell aplasia, resulting in severe immunosuppression and opportunistic infections [102,183,184]. Additionally, the risk of malignant T-cell contamination during CAR-T manufacturing poses a significant safety concern [183].

To overcome these obstacles, CAR-T cells targeting CD7, CD5, and CD1a have been developed, showing antileukemic efficacy and reduced fratricide in preclinical in vitro and in vivo models [184,185]. Particularly promising are fratricide-resistant, universal “off-the-shelf” CD7-targeted CAR-T cells (UCART7), engineered using CRISPR/Cas9 to eliminate CD7 and TCR α-chain expression, thereby preventing self-recognition and graft-versus-host responses [102,186].

A phase I clinical trial (NCT03081910) investigating CD5-targeted CAR-T cells for T-cell malignancies is currently underway, marking an important step toward establishing the feasibility and safety of CAR-T immunotherapy in T-ALL [104].

Tisagenlecleucel (tisa-cel), a second-generation CD19-directed CAR T-cell product, has been approved by the U.S. Food and Drug Administration for pediatric and young adult patients with R/R B-cell ALL, as well as for adults with R/R diffuse large-B-cell lymphoma and R/R follicular lymphoma. Clinical trials have demonstrated a complete remission rate of 82%, although the cumulative incidence of relapse after CD19-directed CAR T-cell therapy remains at 36% [175].

A phase I study investigated CD19/CD22 dual CAR T cells in 17 patients with R/R B-ALL, including 4 previously treated with CD19 CAR T cells. The cohort included both adults and pediatric patients (aged 1–45). The study reported favorable safety and clinical efficacy, with a high CR rate and low toxicity that was dose-dependent. Notably, none of the patients relapsed during a median follow-up of 60 days (range, 7–139) [164].

Compared with traditional chemotherapy, CAR T-cell therapy is generally associated with lower systemic toxicity and is better tolerated. Nevertheless, two major toxicities, CRS and immune effector cell-associated neurotoxicity syndrome (ICANS), are common, particularly in ALL patients [187]. Non-specific side effects such as gastrointestinal symptoms, mucositis, myelosuppression, and constitutional symptoms tend to be mild or absent. However, the risk of severe complications, such as infections, vascular events, severe inflammatory syndromes, and autoimmune manifestations, can be life-threatening depending on the biological target. The reported rates of grade ≥3 adverse events include the following: neutropenia, 38%; thrombocytopenia, 23%; neurotoxicity, 18%; infections, 29%; and CRS, 19% [188].

Despite these risks, CAR-T therapy provides numerous advantages over conventional treatment options. This form of therapy is highly specific and targeted, enabling patients with R/R cancers after multiple prior treatments to achieve complete remissions lasting years. CAR-T cells effectively eliminate cancer cells expressing tumor-associated antigens, unlike conventional adaptive immune cells [180]. Many patients experience prolonged periods without disease progression. Key benefits include the low number of infusions required, a shorter overall treatment period, and faster recovery than traditional therapies. As living cells, CAR-T cells can proliferate within the patient’s body and establish immunological memory that persists for years, allowing them to recognize and kill cancer cells upon relapse. Available data demonstrates durable complete remissions in R/R patients after a single CAR-T infusion [175,187].

Ongoing development of new generations of CAR technologies and their combined application with treatments such as HSCT represent promising approaches for efficiently treating ALL after chemotherapy failure. These advancements have the potential to make CAR-T therapy a safe and effective treatment option for ALL and other hematological malignancies in the future [189].

### 8.1. Short-Term Complications (Day 0 to Day 28)

During the first month after CAR-T cell infusion, patients may develop several acute complications due to the robust immune activation induced by the therapy. These short-term complications typically arise from days to weeks and require prompt identification and management, as they can significantly impact both morbidity and mortality [190]. The most commonly reported early complications include the following:-CRS;-ICANS;-Infections;-Tumor lysis syndrome (TLS);-Hemophagocytic lymphohistiocytosis-like syndrome (HLH).

CRS is the most frequently observed acute toxicity following CAR-T infusion. It results from the massive release of pro-inflammatory cytokines such as IL-6, IFN-γ, and IL-1 upon T-cell activation and target engagement. CRS typically occurs within the first few days after infusion, and its presentation can range from mild flu-like symptoms to severe life-threatening manifestations such as hypotension, hypoxia, and multi-organ dysfunction. Its severity is commonly graded using ASTCT consensus criteria, and treatment often includes supportive care, IL-6 receptor blockade (e.g., tocilizumab), and corticosteroids in severe cases [191].

Closely related to CRS in both timing and pathophysiology, ICANS is a neurotoxicity syndrome characterized by symptoms such as confusion, aphasia, seizures, and, in severe cases, cerebral edema. It generally occurs concurrently with or following CRS but can also occur independently. Although the precise mechanisms remain unclear, endothelial activation, disruption of the blood–brain barrier, and systemic inflammation are key contributors. ICANS is managed with corticosteroids since tocilizumab has limited efficacy due to poor CNS penetration [192].

Infections represent another major concern in the short-term period, as patients are often immunosuppressed due to prior therapies, lymphodepleting chemotherapy, and the effects of CAR-T cell expansion. Bacterial, viral, and fungal infections have all been reported, particularly in the first two weeks post-infusion. Prophylactic antimicrobials, infection surveillance, and prompt antimicrobial therapy are essential components of supportive care during this period [193].

TLS may occur prior to or shortly after CAR-T administration, especially in patients with a high tumor burden. TLS results from the rapid destruction of malignant cells, leading to electrolyte disturbances (hyperuricemia, hyperkalemia, hyperphosphatemia, and hypocalcemia) and the risk of acute kidney injury. Preventive measures such as aggressive hydration, uric acid-lowering agents (e.g., allopurinol or rasburicase), and close biochemical monitoring are critical in high-risk patients [190].

Although rare, a subset of patients may develop an HLH-like syndrome, which shares features with macrophage activation syndrome and may overlap with severe CRS. This complication is marked by sustained fever, elevated ferritin and triglycerides, cytopenias, and evidence of hyperinflammation. Its management typically includes immunosuppression with corticosteroids and may require additional anti-cytokine therapy depending on severity and overlap with CRS [192].

### 8.2. Medium-Term Complications (Day +28 to Day +100)

In the period between day +28 and day +100 after CAR T-cell infusion, patients may experience several medium-term complications, which—although variable in frequency and severity—warrant careful monitoring. These include the following:-Late-onset CRS, ICANS, and TLS;-Opportunistic infections;-Prolonged cytopenias (neutropenia, lymphopenia, and thrombocytopenia);-Hypogammaglobulinemia and B-cell aplasia;-Suboptimal vaccine responses;-Risk of GvHD in post-alloHCT patients.

Although rare, late episodes of CRS, ICANS, or TLS have been observed, and, in late onset, the management strategy is similar to that employed in short-term complications.

Opportunistic infections become a major concern during this phase, particularly as immune reconstitution remains incomplete. Early infections (within 30 days) are usually bacterial or due to respiratory viruses, while later infections are more often viral. Factors that increase infection risk include prior autologous or allogeneic HCT, bridging therapies, and the use of immunosuppressants (like corticosteroids or tocilizumab) during CRS or ICANS [194].

Up to 46% of patients still show hypogammaglobulinemia by day +90, and neutropenia persists in approximately 30% at day +30 and in 10–20% beyond day +90.

Prolonged cytopenias, especially neutropenia and thrombocytopenia, are frequently observed beyond day +30 and may last several months. Their pattern may be monophasic, biphasic, or even triphasic. Among pediatric and young adult patients receiving tisa-cel, 43% had grade 3–4 thrombocytopenia, and 53% had neutropenia persisting past day +30. Contributing factors include intensive lymphodepletion, bridging chemotherapy, bone marrow involvement at baseline, and immune activation (e.g., carHLH) [192].

High levels of IFN-γ, seen in severe CRS, may impair hematopoiesis. Prolonged cytopenias beyond three months (reported in 17–32% of patients) may require bone marrow studies to exclude clonal evolution or myelodysplasia. G-CSF is often recommended from day 14 onward (once CRS/ICANS has resolved), and recent evidence supports its use even in early phases without compromising CAR T-cell expansion or efficacy [192].

Vaccination during this phase remains controversial. Although vaccine efficacy may be suboptimal due to immune suppression or BCA, vaccination is still advised to reduce the risk of preventable infections [195].

Lastly, while GvHD is generally rare after CAR-T therapy—even in patients with prior alloHCT—a few reports, including histologically confirmed GvHD in pediatric recipients of tisa-cel, suggest that it remains a possible complication requiring vigilance [190].

### 8.3. Long-Term Complications: From Day +100

Long-term follow-up (LTFU) beginning from day +100 after CAR-T therapy should be managed by a multidisciplinary team comprising CAR-T physicians, disease-specific specialists, LTFU nursing staff, data managers, and clinical trial personnel. This team is responsible for accurately assessing disease status and identifying late-onset effects.

Common complications observed during this period include prolonged cytopenias, hypogammaglobulinemia, and increased susceptibility to infections, all requiring continuous monitoring and supportive treatment. Neurological complications and pulmonary toxicity, although less frequent, are serious adverse events that significantly increase the risk of mortality in treated patients. Secondary malignancies remain rare but clinically relevant; isolated cases include relapse caused by the accidental transduction of leukemic B cells during manufacturing, as well as a case of myelodysplastic syndrome reported in the ZUMA-1 trial [196].

## 9. Discussion

ALL is the most common pediatric malignancy and is biologically heterogeneous, comprising distinct immunophenotypic subtypes that differ in clinical presentation, genetic alterations, and treatment responses. The two major categories are B-ALL and T-ALL [1,2,3,4,5]. B-ALL accounts for approximately 85% of pediatric cases and is characterized by a variety of genetic abnormalities such as the Philadelphia chromosome, *CRLF2* rearrangements, and *KMT2A* rearrangements, each influencing prognosis and therapeutic approaches [10,16]. T-ALL, comprising about 15% of cases, often presents with higher leukocyte counts, mediastinal masses, and distinct molecular drivers, including *NOTCH1* mutations and alterations in cell cycle regulation pathways [17,19]. The differentiation between these subtypes is essential for risk stratification and treatment tailoring.

The treatment of ALL includes four phases: induction, consolidation, re-induction, and maintenance therapy [34]. Induction aims to achieve rapid remission with intensive chemotherapy regimens, including corticosteroids, vincristine, anthracycline, asparaginase, cyclophosphamide, and cytarabine. Consolidation intensifies treatment to eradicate MRD, employing agents like HD-MTX for SR and MR patients and blocks of polychemotherapy together with immunotherapy for HR patients. Re-induction is based on the same drug administration employed in the induction phase. Maintenance therapy involves prolonged administration of lower-intensity agents to sustain remission [31,32,33,34,35,36].

Despite advances in chemotherapy protocols leading to improved survival rates, conventional treatments are associated with significant toxicities affecting multiple organ systems [37]. Mucositis, a frequent complication particularly occurring after the administration of high-dose antimetabolites and alkylating agents during induction and consolidation, disrupts mucosal integrity, predisposing patients to infections [37,38,39]. Skeletal toxicities such as OP and ON, driven mainly by corticosteroids and disease-related factors, cause morbidity and long-term disability [40,41,42,43,44,45]. Endocrine dysfunctions like growth impairment, hyperglycemia, and adrenal insufficiency are common sequelae of corticosteroid and asparaginase exposure [56,57,58,59,60,61,62,63,64,65,66]. Other serious complications include SOS and TE, complicating intensive therapy phases [72,73,74,75]. Hepatotoxicity is also a well-recognized adverse effect, mainly occurring during maintenance therapy and related to metabolites of 6-MP and MTX; severe forms such as SOS and NRH are less frequent but clinically relevant [67,68,69,70,71].

The advent of targeted therapies represents a crucial change in the management of ALL by exploiting molecular vulnerabilities that are specific to subtypes and genetic alterations. TKIs such as imatinib, dasatinib, and ponatinib have substantially improved outcomes in Ph+ and Ph-like B-ALL by selectively inhibiting aberrant kinase activity during induction and consolidation. However, these agents can cause off-target toxicities, including cardiac and endocrine adverse effects [32,89,90,91,93,94,95].

Additional targeted therapies target different pathways commonly dysregulated in both B-ALL and T-ALL. For example, JAK-STAT pathway inhibitors (e.g., ruxolitinib) have demonstrated efficacy in *CRLF2*-rearranged Ph-like B-ALL but require combination with other therapeutic approaches due to resistance observed with monotherapy [96]. PI3K/AKT/mTOR inhibitors (e.g., everolimus) show promise in overcoming chemotherapy resistance, while BCL-2 family inhibitors—such as venetoclax—target anti-apoptotic mechanisms frequently upregulated in both subtypes. Emerging agents targeting epigenetic regulators (menin inhibitors for KMT2A rearrangements), proteasome inhibitors, MEK inhibitors (for RAS pathway mutations common in both B-ALL and T-ALL), and CDK4/6 inhibitors (notably in T-ALL with NOTCH1-driven cyclin D3 overexpression) are expanding the therapeutic landscape [114,115,116,117,118,119].

Including these molecular targeted agents in traditional chemotherapy regimens has the potential to enhance efficacy and mitigate toxic side effects by enabling reduced dosages or overcoming resistance. However, targeted therapies have their own side effects, such as hematologic suppression and increased thrombosis risk, which require careful patient monitoring and individualized treatment strategies based on immunophenotype and genetic profiling.

Beyond small-molecule targeted agents, immunotherapy has emerged as a pivotal strategy in pediatric ALL, aiming to enhance anti-leukemic efficacy while reducing chemotherapy intensity.

Blinatumomab, a bispecific T-cell engager (BiTE^®^) antibody construct, simultaneously binds CD3 on T cells and CD19 on B cells, redirecting T-cell cytotoxicity toward leukemic precursors. Its continuous intravenous infusion ensures sustained engagement, leading to rapid B-cell depletion and effective MRD eradication. Clinical trials such as NCT01471782, RIALTO (NCT02187354), and AALL1331 (NCT02101853) demonstrated superior MRD-negative remission and improved EFS with fewer severe toxicities than conventional chemotherapy. Recent frontline studies (e.g., Interfant-21, AIEOP-BFM ALL 2017) confirmed its efficacy as post-induction or consolidation therapy, with mild neurological events as the main adverse effects. Ongoing trials are extending its use to Ph+ and Ph-like B-ALL, and a subcutaneous formulation is under development to simplify administration. Overall, blinatumomab represents a major advancement in replacing intensive chemotherapy blocks with effective, targeted immunotherapy in pediatric B-ALL [127,128,129,130,131,132,133,134,135,136,137,138,139,140,142,143,144,146,147,148,149,150,151,152,197,198,199,200].

In addition to B-ALL, emerging therapeutic strategies are being investigated in T-ALL, particularly for R/R cases. CAR-T-cell therapy targeting CD5, CD7, and CD1a has shown promising preclinical efficacy while addressing fratricide and T-cell aplasia [102,183,184,185,186]. Daratumumab, targeting CD38, has demonstrated activity in patient-derived T-ALL models and is currently being evaluated in early-phase clinical trials in combination with standard chemotherapy [104,155,156,157,158,159]. Targeted approaches, including BH3 mimetics (e.g., venetoclax) for BCL-2-dependent T-ALL subtypes [107,108,109], mTOR inhibitors (e.g., everolimus) for PI3K/Akt/mTOR pathway activation [102,103], and JAK/STAT inhibitors for *IL7R*/*JAK*/*STAT*-mutated T-ALL [97,98,99,100], are expanding the therapeutic landscape and may be integrated with conventional regimens to optimize efficacy while minimizing toxicity [104,111,112,113].

Daratumumab, a monoclonal antibody targeting CD38, has shown promising preclinical and early clinical activity in pediatric hematologic malignancies, including T-ALL and lymphoblastic lymphoma. By inducing antibody- and complement-dependent cytotoxicity and modulating immune cell signaling, daratumumab enhances leukemic cell apoptosis. A phase I–II trial combining daratumumab with backbone chemotherapy indicated potential as a bridge to HSCT in R/R T-ALL. The treatment is generally well tolerated, with infusion-related reactions, fatigue, and mild infections as the most frequent adverse events [88,154,173].

InO, an anti-CD22 monoclonal antibody–drug conjugate linked to calicheamicin, has been recently approved by the FDA for pediatric patients aged ≥1 year with R/R CD22^+^ B-ALL. It induces apoptosis following internalization and intracellular drug release. The ITCC-059/AALL1621 phase I/II study (NCT02981628) reported complete remission in ~80% of treated children, with high MRD clearance rates. InO is particularly useful as a bridge to HSCT or CAR-T-cell therapy but carries a notable risk of hepatic VOD, especially when transplantation follows shortly after therapy. Current trials are evaluating its use as a reinduction or consolidation alternative to standard chemotherapy in first-relapse B-ALL [161,162,163,164,165,166,167,168,169,170,171,172,173,174,175,176,178].

Finally, CAR-T-cell therapy has emerged as a novel promising therapeutic strategy for R/R B-ALL. CAR-T cells are genetically modified T lymphocytes designed to recognize tumor-associated antigens independently of HLA presentation. Second-generation CAR-T products, such as CD19-targeting tisagenlecleucel, have demonstrated high complete remission rates in pediatric R/R B-ALL, with a favorable toxicity profile. This approach not only overcomes chemotherapy resistance but also enables long-term disease control through immune memory formation [179].

The integration of targeted and immune-based therapies into conventional chemotherapy protocols marks a decisive shift toward precision medicine in pediatric ALL. This approach allows treatment intensity to be modulated according to the patient’s molecular and immunophenotypic profile, aiming to maximize efficacy while minimizing cumulative toxicity. The incorporation of TKIs, monoclonal antibodies, bispecific engagers, and CAR-T cells has already demonstrated the potential to replace or reduce highly toxic chemotherapy blocks, particularly in relapsed or high-risk subgroups. Moreover, this strategy supports a paradigm in which therapy is increasingly guided by MRD dynamics and genomic biomarkers rather than by broad risk categories. As a result, survival outcomes are improving, while treatment-related morbidity and long-term sequelae—such as endocrine dysfunction, bone toxicity, and secondary malignancies—are progressively decreasing. Therefore, the convergence of targeted and immune-based therapies into conventional treatment protocols represents a critical step toward personalized, risk-adapted treatment and improved quality of life for children with ALL.

Overall, the integration of immunophenotypic classification with molecular and genetic insights is crucial to optimize treatment strategies across all therapeutic phases—from induction to maintenance. The incorporation of personalized, patient-tailored approaches, including targeted therapies and immunotherapeutic strategies such as CAR-T-cell therapy, in combination with standard chemotherapy regimens, enhances the potential to improve survival outcomes and quality of life in both pediatric B-ALL and T-ALL patients [180,181,182,187,188,189,190,191,192,193,194,195,196,200].

## 10. Conclusions

The understanding of the molecular and genetic profiles of pediatric ALL has significantly affected the knowledge of treatments, allowing for specific patient stratification and personalized therapy. Obviously, the distinction between B- and T-ALL subtypes, each with specific biological characteristics and prognostic implications, is important for developing targeted therapeutic approaches. Conventional multi-phase chemotherapy protocols—including induction, consolidation, re-induction, and maintenance—continue to represent the main therapeutic therapies to counteract ALL progression, leading to increased remission rates.

However, conventional anti-cancer therapies entail many short- and long-term side effects, and, therefore, the search for new therapeutic strategies that reduce the side effects associated with conventional therapies is necessary. The combination of targeted therapies, such as TKIs, JAK-STAT pathway inhibitors, BCL-2 antagonists, and emerging agents against epigenetic and cell cycle regulators, enhances treatment efficacy and overcomes drug resistance.

In this context, immunotherapeutic agents such as blinatumomab (a bispecific T-cell engager targeting CD19), InO (an anti-CD22 antibody–drug conjugate), and daratumumab (an anti-CD38 monoclonal antibody) have shown remarkable results in R/R B-ALL, offering new perspectives in disease management. Their integration into treatment protocols, either alone or in combination with conventional chemotherapy, has the potential to improve outcomes while minimizing toxicity, marking an important step toward more precise and less harmful therapeutic strategies for pediatric ALL.

In parallel, CAR-T-cell therapy has emerged as a powerful immunotherapeutic option, particularly for R/R B-ALL, offering high remission rates with reduced systemic toxicity. Targeted therapies, when administered properly in combination with conventional chemotherapy, can reduce chemotherapy-related toxicity and promote a greater effect by targeting specific molecular alterations in ALL, as well as when combined with CAR-T-cell therapy in selected high-risk or relapsed cases.

Ongoing clinical trials and translational research are essential to validate the long-term benefits and safety profiles of these novel agents, especially in the pediatric population. In conclusion, the main aim is to improve treatments minimizing acute and late toxicities, enhancing both survival and quality of life for ALL children.

## Figures and Tables

**Figure 1 ijms-26-11362-f001:**
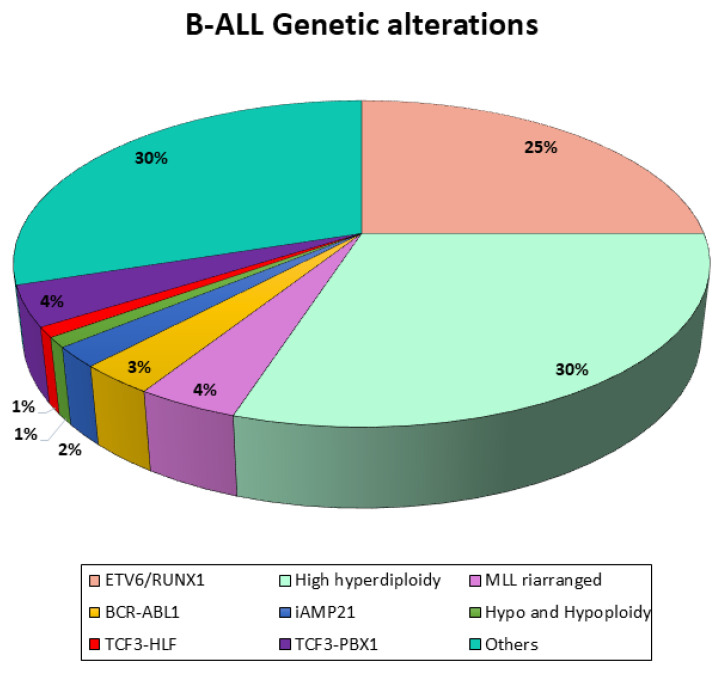
Genetic landscape of B-ALL. Schematic representation of genetic alterations in B-ALL.

**Figure 2 ijms-26-11362-f002:**
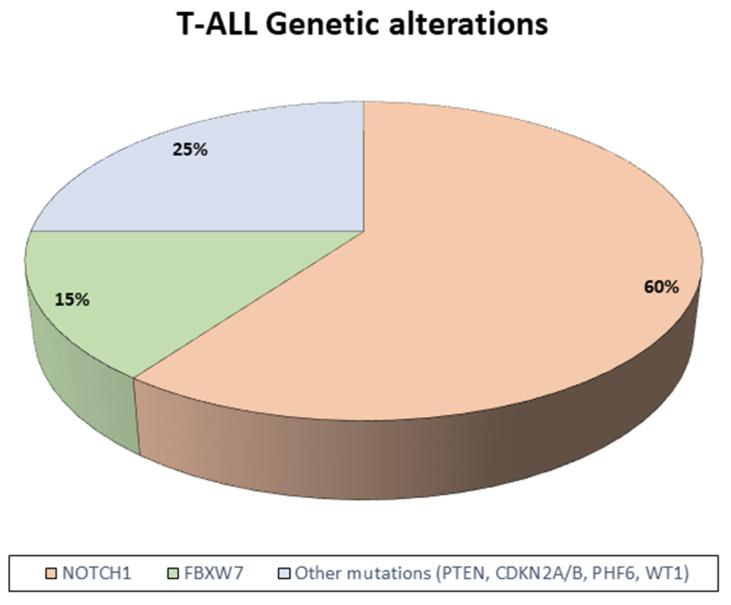
Genetic landscape of T-ALL. Schematic representation of genetic alterations in T-ALL.

**Figure 3 ijms-26-11362-f003:**
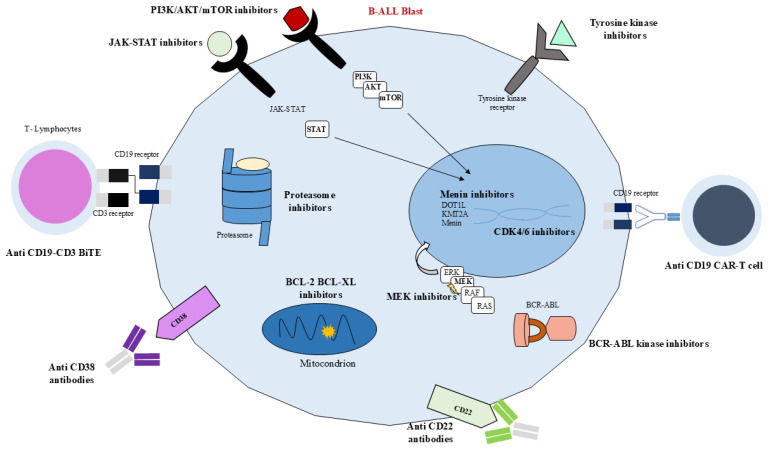
Biological targets in B-ALL. JAK-STAT inhibitors, PI3K/AKT/mTOR inhibitors, tyrosine kinase inhibitors, CD19-CD3 BiTE, anti-CD38 antibodies, anti-CD22 antibodies, BCR-ABL kinase inhibitors, anti-CD19 CAR-T cells, Ras-Raf-MEK-ERK inhibitors, proteasome inhibitors, BCL-2-BCL-XL inhibitors, menin inhibitors, CDK4/6 inhibitors.

**Figure 4 ijms-26-11362-f004:**
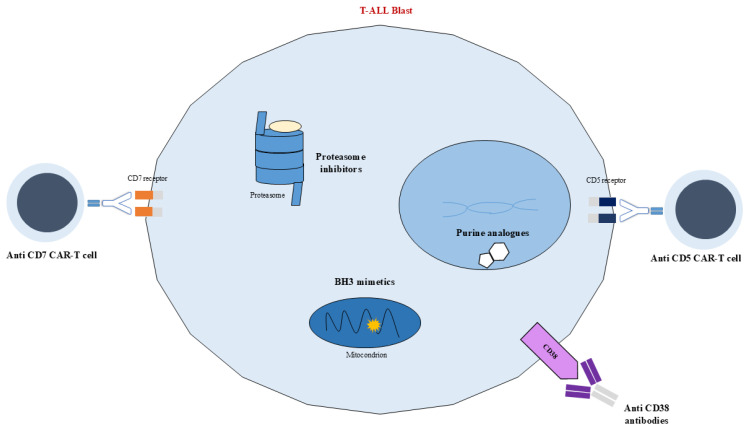
Schematic representation of targeted therapies. Biological targets in T-ALL. Anti-CD7 CAR-T cell, anti-CD38 antibodies, BH3 mimetics, proteasome inhibitor, anti-CD5 CAR-T cell.

**Figure 5 ijms-26-11362-f005:**
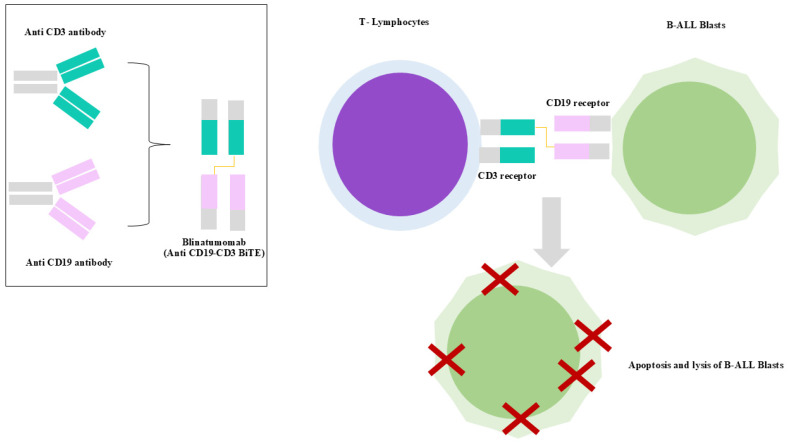
Mechanism of action of blinatumomab. The bispecific T-cell engager simultaneously binds to CD19 on B-ALL blasts and CD3 on T cells, redirecting cytotoxic T-cell activity to induce apoptosis of malignant CD19+ blasts.

**Figure 6 ijms-26-11362-f006:**
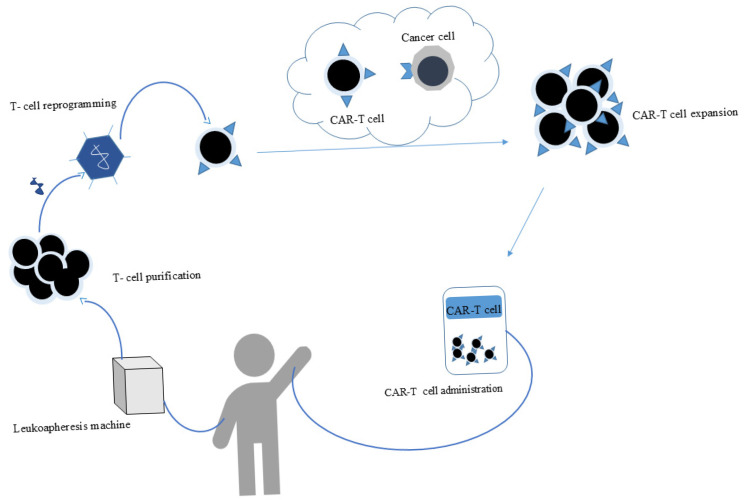
Schematic representation of CAR-T-cell therapy. Patient T cells are collected through leukapheresis and genetically engineered to express a chimeric antigen receptor (CAR) that specifically recognizes a tumor-associated antigen (e.g., CD19). After expansion, CAR-T cells are reinfused into the patient, where they recognize and kill malignant cells through targeted immune activation.

## Data Availability

No new data were created or analyzed in this study. Data sharing is not applicable to this article.

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
