# Peer review of "Novel Therapeutic Approaches in Pediatric Acute Lymphoblastic Leukemia"

_ijms, 2025, doi:10.3390/ijms262311362_

Round 1
Reviewer 1 Report
Comments and Suggestions for Authors
The underlying review is very interesting and well written. The authors present a detailed overview of the pathobiology of acute lymphoblastic leukaemia and a review of the current therapeutic options for the disease. I just have a few suggestions and minor comments. In my opinion it is worth mentioning and describing hepatotoxicity as a common adverse drug reaction during conventional treatment of ALL.I would add the Ras-Raf-MEK-ERK pathway in the title of figure 3, as it is shown there and described in the text.Chapter 7.2. gives less information on Daratumomabab than the other chapters regarding Immunotherapeutics. Especially pharmacokinetic and safety data is limited.In my opinion it would be important to mention QT prolongation as an adverse drug reaction of Inotuzumab ozogamicin.Page 17 line 725. A reference should be given.Page 23 line 976 – adverse drug reaction or side effect. I would not write "adverse side effects".
Author Response
Reviewer 1
Comments and Suggestions for Authors
The underlying review is very interesting and well written. The authors present a detailed overview of the pathobiology of acute lymphoblastic leukaemia and a review of the current therapeutic options for the disease. I just have a few suggestions and minor comments. In my opinion it is worth mentioning and describing hepatotoxicity as a common adverse drug reaction during conventional treatment of ALL. I would add the Ras-Raf-MEK-ERK pathway in the title of figure 3, as it is shown there and described in the text. Chapter 7.2. gives less information on Daratumomabab than the other chapters regarding Immunotherapeutics. Especially pharmacokinetic and safety data is limited. In my opinion it would be important to mention QT prolongation as an adverse drug reaction of Inotuzumab ozogamicin. Page 17 line 725. A reference should be given. Page 23 line 976 – adverse drug reaction or side effect. I would not write "adverse side effects".
Response to Reviewer 1
We thank the reviewer for the valuable comments and constructive suggestions to improve the quality of our manuscript. All comments have been addressed as detailed below:
- Hepatotoxicity during conventional ALL treatment: We have added in paragraph “5. Toxicity related to conventional therapy”, under the subparagraph “5.5 Hepatotoxicity”, a discussion on hepatotoxicity as a common adverse drug reaction during conventional treatment of acute lymphoblastic leukemia.
- Figure 3 - Ras-Raf-MEK-ERK pathway: The title of Figure 3 has been modified to include the Ras-Raf-MEK-ERK pathway, consistent with its depiction and discussion in the text.
- Section 7.2 - Daratumumab (pharmacokinetic and safety data): In the subparagraph “7.2. Daratumomab”, we include pharmacokinetic and safety data. Specifically, we added information on concentration-response relationships, trough concentration maintenance, and data from previous population pharmacokinetic analyses in multiple myeloma.
- QT prolongation as an adverse drug reaction of Inotuzumab ozogamicin: A section describing QT prolongation as a possible adverse drug reaction, supported by clinical data and pharmacovigilance reports, has been added to the subparagraph “7.3. Inotuzumab ozogamicin”.
- Page 17, line 725 – Reference: A reference has been added to support the statement mentioned in this line.
- Page 23, line 976 - Terminology correction: The term “adverse side effects” has been corrected to “side effects” to ensure terminological accuracy.
All corresponding changes are highlighted in the revised version of the manuscript.
Reviewer 2 Report
Comments and Suggestions for Authors
Novel therapeutic approaches in pediatric acute lymphoblastic leukemia by Marrapodi MM et al.
The manuscript outlines the genetic alterations to distinguish B-cell and T-cell ALL, as well as cytofluorimetric markers for the differential diagnosis of these diseases. Then, the authors describe the effect of conventional therapy, which significantly increased the survival rates in ALL patients, although it was accompanied by multiple side effects. Indeed, the review is interesting and detailed, with information that could be discussed and tied to each other, and not just a list. This would enhance the attraction of the wide readership of the Journal. The authors are very competent on the topic; therefore, the manuscript is particularly extended, but the real focus of the review is unclear to the reader. It could also be better highlighted in the Abstract. This could make it more attractive.
Major Comments:
Although the authors reports that, lines 1030-32 page 24: “Obviously, the distinction between B- and T-ALL subtypes, each with specific biological characteristics and prognostic implications, is important for developing targeted therapeutic approaches.” the review is dedicated to B-ALL and only few aspects of T-ALL are considered, essentially, in the therapy section nor there is any comparison between the two. Some comments to the authors could be suggested in the section on Target Therapy. First, Figure 3 is completely dedicated to B-ALL. Regarding CAR T, which has been proven effective in B-ALL, its effectiveness in T-ALL is still under study. For what BH3 mimetics, there are some differences between ETP-ALL and non-ETP ALL. The authors are suggested to discuss these points.
Lines 33-35 page. Abstract: “The integration of targeted and immune-based therapies into conventional regimens represents a decisive step toward precision medicine, aiming to enhance survival outcomes while reducing treatment-related toxicity and improving quality of life in ALL children. This is interesting, and its discussion is suggested to be improved to strengthen the content.
Minor comments
Lines 201-213 page 5. These two paragraphs would benefit from the integration of more recent and updated articles. All the notions seem to be poorly referenced. Potential suggestions are PMID 36674902 and 32072504.
Figure 4. This figure would benefit from a better combination of colours to better understand the message.
Lines 526-527, page 13. The sentence is not clear.
Line 532, the number “12 Nevertheles …” is appropriate? Perhaps the references are not inserted properly.
Author Response
Reviewer 2
Comments and Suggestions for Authors
Novel therapeutic approaches in pediatric acute lymphoblastic leukemia by Marrapodi MM et al.
The manuscript outlines the genetic alterations to distinguish B-cell and T-cell ALL, as well as cytofluorimetric markers for the differential diagnosis of these diseases. Then, the authors describe the effect of conventional therapy, which significantly increased the survival rates in ALL patients, although it was accompanied by multiple side effects. Indeed, the review is interesting and detailed, with information that could be discussed and tied to each other, and not just a list. This would enhance the attraction of the wide readership of the Journal. The authors are very competent on the topic; therefore, the manuscript is particularly extended, but the real focus of the review is unclear to the reader. It could also be better highlighted in the Abstract. This could make it more attractive.
Major Comments:
Although the authors reports that, lines 1030-32 page 24: “Obviously, the distinction between B- and T-ALL subtypes, each with specific biological characteristics and prognostic implications, is important for developing targeted therapeutic approaches.” the review is dedicated to B-ALL and only few aspects of T-ALL are considered, essentially, in the therapy section nor there is any comparison between the two. Some comments to the authors could be suggested in the section on Target Therapy. First, Figure 3 is completely dedicated to B-ALL. Regarding CAR T, which has been proven effective in B-ALL, its effectiveness in T-ALL is still under study. For what BH3 mimetics, there are some differences between ETP-ALL and non-ETP ALL. The authors are suggested to discuss these points.
Lines 33-35 page. Abstract: “The integration of targeted and immune-based therapies into conventional regimens represents a decisive step toward precision medicine, aiming to enhance survival outcomes while reducing treatment-related toxicity and improving quality of life in ALL children. This is interesting, and its discussion is suggested to be improved to strengthen the content.
Minor comments
Lines 201-213 page 5. These two paragraphs would benefit from the integration of more recent and updated articles. All the notions seem to be poorly referenced. Potential suggestions are PMID 36674902 and 32072504.
Figure 4. This figure would benefit from a better combination of colours to better understand the message.
Lines 526-527, page 13. The sentence is not clear.
Line 532, the number “12 Nevertheles …” is appropriate? Perhaps the references are not inserted properly.
Response to Reviewer 2
We thank the Reviewer for the constructive comments to improve our manuscript. Below, we provide a detailed point-by-point response.
In the revised version, we have modified the Abstract by adding a brief section that clearly highlights the aim and focus of the review. Moreover, we expanded the Discussion section to better address the concept expressed in Lines 33-35 page of Abstract. Particularly, we now emphasize how the combination of targeted and immune-based therapies with standard chemotherapy regimens represents a major step toward precision medicine, improving survival while reducing toxicity and long-term sequelae.
Major Comments
We have substantially expanded the sections dedicated to T-ALL, particularly in the Targeted Therapies and Immunotherapy sections, and also, we have better discussed CAR-T cell therapy in T-ALL and BH3 mimetics, as suggested. xpanded the Discussion to better address this concept.
We now emphasize how the combination of targeted and immune-based therapies with standard chemotherapy regimens represents a major step toward precision medicine, improving survival while reducing toxicity and long-term sequelae.
We have added another figure (Figure 4) to include T-ALL related elements, making it more balanced and representative of both B-ALL and T-ALL pathways. Consequently, the numbering of the subsequent figures has been updated throughout the manuscript.
Minor Comments
Lines 201-213 page 5. We have revised this section and integrated the suggested reference (PMID: 36674902).
Figure 4. We have improved this figure (now Figure 5) with better combination of colours to better understand the message.
Lines 526-527, page 13. We have rewritten the sentence for clarity and readability.
Line 532. The incorrect reference (12) has been removed, and the citation list has been verified and corrected accordingly.
Round 2
Reviewer 2 Report
Comments and Suggestions for Authors
All the requested modifications have been performed.